# GeoFAR: Geography-Informed Frequency-Aware Super-Resolution for Climate Data

**Chang Xu**[1*]**, Gencer Sumbul**[1]**, Li Mi**[1,2]**, Robin Zbinden**[1]**, Devis Tuia**[1]
[1]EPFL, Switzerland    [2]ETH Zurich, Switzerland

## Abstract

Super-resolving climate data is crucial for fine-grained decision-making in various domains, ranging from agriculture to environmental conservation. However, existing super-resolution approaches struggle to generate the high-frequency spatial information present in climate data, especially over regions showing complex terrain variability. A key obstacle lies in a frequency bias existing in both deep neural networks (DNNs) and climate data: DNNs exhibit such bias by overfitting to low-frequency information, which is further exacerbated by the prevalence of low-frequency components in climate data (*e.g.*, plains, oceans). As a consequence, geography-dependent high-frequency details are hard to reconstruct from coarse climate inputs with DNNs. To improve the fidelity of climate super-resolution (SR), we introduce GeoFAR: by explicitly encoding climatic patterns at different frequencies, while learning implicit geographical neural representations (*i.e.*, related to location and elevation), our approach provides frequency-aware and geography-informed representations for climate SR, thereby reconstructing fine-grained climate information at high resolution. Experiments show that GeoFAR is a model-agnostic approach that can mitigate high-frequency prediction errors in both deterministic and generative SR models, demonstrating state-of-the-art performance across various spatial resolutions, atmospheric variables, and downscaling ratios. Datasets and code are available at: `https://eceo-epfl.github.io/GeoFAR/`.

## 1 Introduction

The inherent complexity of the climate system leads to complex regional climate variations at local scales. As an example, NOAA's HeatWatch campaigns show intra-urban air-temperature differences of up to 9°C in records only a few kilometers apart, mostly due to terrain and ventilation variability (NOAA Climate Program Office, 2025). Fine-grained, accurate climate observations or estimations are thus crucial for site-specific decision-making in areas as diverse as agriculture, environmental conservation, and hydrological management.

Climate downscaling provides a way to obtain such fine-scale climate details from coarse inputs by either physics-based dynamical methods or data-driven statistical methods (Sun et al., 2024). With respect to the latter, climate downscaling has come to be formulated as a super-resolution (SR) task with deep learning methods (Vandal et al., 2017; Stengel et al., 2020; Baño-Medina et al., 2022). Compared to physics-based dynamical models, deep neural networks (DNNs) achieve competitive performance with much lower computational cost (Lopez-Gomez et al., 2025), making image SR approaches an effective solution for climate downscaling.

However, the loss of high-frequency details remains a challenge in image SR (Jiang et al., 2021; Wang et al., 2020), and this is exacerbated in climate SR. Compared to natural images (Martin et al., 2001), climate data is severely biased towards low-frequency components, as shown in the comparison of frequency distributions in Figure 1a. The frequency distribution in climate data is also location- and elevation-dependent: plains are dominated by low-frequency variations, whereas mountainous regions contain richer high-frequency content. The inherent frequency bias of deep neural networks (Rahaman et al., 2019) is further exacerbated by this geographical bias in climate

---

*Correspondence: chang.xu@epfl.ch

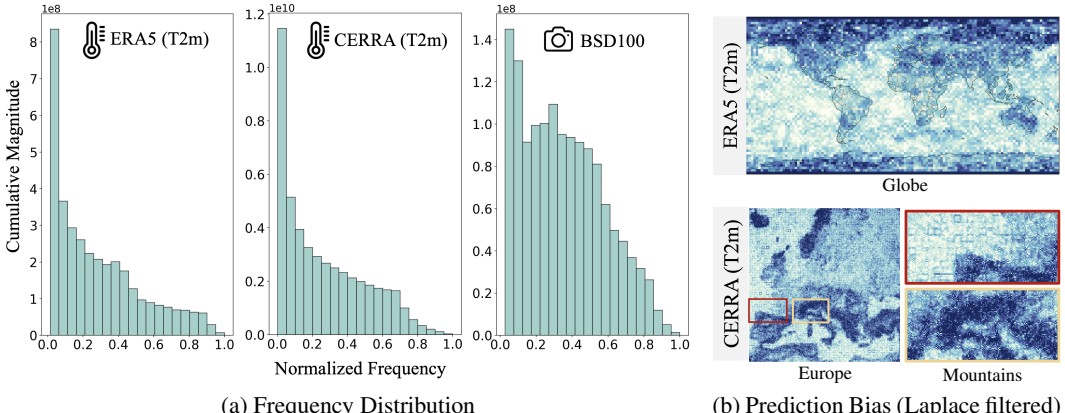

(a) Frequency Distribution     (b) Prediction Bias (Laplace filtered)

Figure 1: **Frequency bias in climate super-resolution**. (a) Climate data (ERA5, CERRA) contains much higher low-frequency information compared to natural images (BSD100) (x-axis: the radial spatial frequency scaled to [0,1]; y-axis: the sum of Fourier frequency magnitudes in each bin). (b) An example of SR results shows geography-dependent high-frequency errors: the Laplace filtered prediction bias highlights errors over polar regions (Globe), coastlines (Europe), and mountains.

SR: DNNs are prone to fit the large amount of smoothly changing regions (e.g, ocean and plains), and fail to reproduce the high-frequency climate details, usually associated with complex terrain variability. Such bias toward low frequencies leads to over-smoothed or hallucinated regional estimations (see Appendix A.8). This behavior of geography-dependent high-frequency loss is illustrated in Figure 1b: the Laplace filtered prediction bias (where darker blue denotes a larger high-frequency deficit) reveals a concentrated high-frequency error over polar regions, coastlines, and mountains.

To address the above-mentioned challenges in climate SR, we introduce a Geography-Informed and Frequency-Aware Super-Resolution (GeoFAR) approach for high-fidelity climate downscaling. GeoFAR mitigates the low-frequency aggregation in climate data by learning fine-grained Frequency-Aware Representations (FAR) with frequency-aware convolution kernels that explicitly encode both low-frequency components and high-frequency details. In addition, GeoFAR learns implicit neural representations for geography (Geo-INR) specific to climate SR. Geo-INR goes beyond location-only encoding in prior studies (Rußwurm et al., 2024; Mai et al., 2020) by jointly encoding location and elevation (terrain)-specific implicit representations to inform climate SR, thereby capturing the dependency of climate states on fine-grained geographical characteristics.

Experiments on three heterogeneous climate downscaling datasets (ERA5, PRISM, and the proposed CERRA high-resolution datasets) show that GeoFAR learns frequency-aware and geography-informed representations to accurately reconstruct local-scale climate information. Our proposed geography-informed learning (Geo-INR) outperforms baselines that stack elevation as an additional channel to the atmospheric variables for SR. Across both deterministic and generative SR baselines, GeoFAR significantly reduces high-frequency prediction errors and achieves state-of-the-art performance: 1) across global (ERA5), global-to-local (ERA5→PRISM), and local downscaling (CERRA) settings (Table 1); 2) on both surface (*e.g.*, 2m-temperature) and pressure-level variables (*e.g.*, geopotential-500hPa, temperature-850hPa) (Table 3a); and 3) of 44→5.5km resolution gains with errors below half a unit (Table 3b). In summary, our contributions are:

- We propose a Geography-Informed and Frequency-Aware (GeoFAR) approach for high-fidelity climate SR: GeoFAR learns fine-grained frequency-aware representations for climate data and modulates these representations with geographical implicit neural representations.

- GeoFAR effectively tackles the geography-dependent frequency bias in climate SR: by mitigating low-frequency aggregation and reconstructing geography-related high-frequency climatic details, GeoFAR yields significant improvements in regions with complex terrain.

- Experiments on reanalysis (ERA5, CERRA) and observational (PRISM) climate data demonstrate GeoFAR's adaptability to both deterministic and generative baselines, achieving state-of-the-art performance across diverse spatial resolutions, atmospheric variables, and downscaling ratios.

## 2 RELATED WORK

### 2.1 CLIMATE DOWNSCALING

Traditional climate downscaling methods simulate regional climate from global climate models. This is referred as *dynamical downscaling* (Tapiador et al., 2020; Sun et al., 2024). While grounded in physical principles, dynamical downscaling is computationally expensive and inherits the biases of the global models. Data-driven *statistical downscaling* (Sun et al., 2024) has achieved competitive accuracy at much lower computational cost with growing adoption of DNNs inspired by image super-resolution (SR) (Dong et al., 2016; Lim et al., 2017; Liang et al., 2021). We categorize these SR-based approaches into deterministic and generative models. *Deterministic models* learn a single mapping from coarse-scale to higher-resolution climate data by minimizing the loss at the pixel level. Early works, such as DeepSD (Vandal et al., 2017), utilize cascade convolutional neural network (CNN) blocks with concatenated climate and topographic inputs. More recent methods further extend DeepSD to downscale multi-modal ensembles (Baño-Medina et al., 2022) or integrate atmospheric processes as constraints (de Roda Husman et al., 2024; Chen et al., 2022). However, deterministic models tend to smooth the high-frequency details and do not model prediction uncertainty. *Generative models* address these challenges by sampling from a distribution conditioned on the low-resolution image, therefore learning to represent fine-scale details and capture local variations and extremes. Previous efforts have applied Generative Adversarial Networks (GANs, (Goodfellow et al., 2020)) for physics-informed climate downscaling (Stengel et al., 2020; Iotti et al., 2025; Li & Cao, 2025; Lopez-Gomez et al., 2025). More recently, diffusion-based models (Liu & Tang, 2025; Ho et al., 2020) have been adapted to iteratively denoise coarse inputs, surpassing deterministic models in fine-scale variance and offering probabilistic outputs for climate downscaling (Watt & Mansfield, 2024; Srivastava et al., 2024). Despite this progress, existing studies do not explicitly investigate the geography-dependent frequency characteristics of climate downscaling, leading to a limited utilization of the (geographic) domain knowledge to modulate high-frequency content in complex terrains, and therefore often resulting in oversmoothed SR outputs.

### 2.2 FREQUENCY-AWARE MODELING

Every signal (from audio, images, to videos) exhibits a combination of high and low frequency components, while DNNs are known to be biased towards low frequency components: they converge well on low-frequency components but struggle with the high-frequency ones (Schwarz et al., 2021; Tancik et al., 2020; Xu et al., 2020). This has motivated a line of work on improving high-frequency modeling for both generic architectures and task-specific settings. Generic methods, including periodic activation functions (Sitzmann et al., 2020), Fourier Neural Operators (Li et al., 2021), and frequency-aware Vision Transformers (Bai et al., 2022; Lee et al., 2025), aim to enhance the representation of high-frequency information regardless of the downstream task. Task-specific frequency-aware methods (e.g., using Wavelet Transform (Kingsbury, 2001)) are proposed to enhance high-frequency components (Jiang et al., 2021; Wu et al., 2023) in image reconstruction, eliminate small artifacts in SR (Korkmaz et al., 2024; Fuoli et al., 2021; Kim et al., 2025), or compress the model with frequency-aware models (Xie et al., 2021). Another line of frequency-aware modeling introduces attention mechanisms to enhance specific frequency components. Frequency attention first decouples channels into different frequency bands, and then learns frequency-aware reweighting (Qin et al., 2021; Ulicny et al., 2022; Chen et al., 2024). However, classic wavelet-based approaches with four frequency bands tend to provide limited separation of high-frequency content in climate data, whose distribution is concentrated at low frequencies. On the contrary, learnable frequency attention is mainly designed for recognition tasks (e.g., classification) and can further exacerbate the loss of high-frequency information due to its frequency truncation and the neural network's inherent bias towards low frequencies. A fine-grained and unbiased decoupling method tailored for climate data is still missing.

### 2.3 GEOGRAPHY-INFORMED LEARNING

Climatic processes are strongly shaped by geography, since sharp changes in location and terrain can produce pronounced local climatic responses (Pepin & Lundquist, 2008). Consequently, it is important for climate models to incorporate geography (i.e. location and elevation information) to improve their physical realism, internal consistency, and predictive performance. In the context of climate

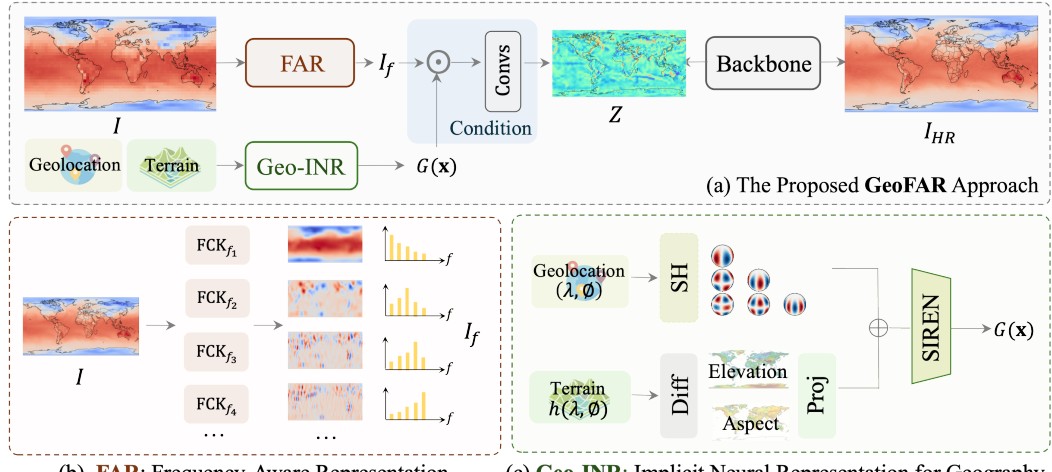

Figure 2: **An overview of GeoFAR.** (a) Model pipeline, which can be directly plugged into deterministic models or used as the generator in generative models. (b) The frequency-aware representation decomposes the image into frequency subbands with frequency-aware convolution kernels (FCK). (c) Geo-INR encodes both location and terrain information for geography-informed SR.

downscaling, classic methods usually incorporate location as geographically weighted regression to enhance performance (Zhao et al., 2017), and topography is sometimes used as an additional predictor (Fiddes & Gruber, 2014). For example, recent deep learning-based climate downscaling approaches have included elevation as an additional channel input to CNNs (Vandal et al., 2017). However, simply stacking elevation alongside climate variables is suboptimal: the model has to process together fundamentally different data types. A growing line of research is exploring how to better encode geographic priors into deep learning models for a wide range of geospatial tasks (*e.g.*, species distribution modeling (Cole et al., 2023), satellite image classification (Ayush et al., 2021), weather forecasting (Verma et al., 2024)). These works project geographic locations into implicit neural representations to more faithfully capture spatial relationships and better inform the downstream task (Mai et al., 2020; Rußwurm et al., 2024). However, geographic implicit representations remain underexplored in climate downscaling: climatic states vary significantly with both location and elevation, making it essential to go beyond current single-factor methods (conditioning on location *or* elevation) and rather jointly condition the super-resolution process on both.

## 3    GEOGRAPHY-INFORMED FREQUENCY-AWARE SUPER-RESOLUTION

Let an atmospheric variable be defined on a grid $I \in \mathbb{R}^{H \times W}$ for representation learning with neural networks (Nguyen et al., 2023), where $H \times W$ is the spatial extent, either defined for the whole globe or for a specific region at a given resolution. Climate downscaling seeks to provide a high-resolution output $I_{\text{HR}} \in \mathbb{R}^{H' \times W'}$ conditioned on the input $I$.

To enhance the fidelity of climate downscaling, GeoFAR aims to learn geography-informed and frequency-aware climate representations $Z \in \mathbb{R}^{d \times H \times W}$ specific to SR, where each spatial location is associated with a $d$-dimensional feature vector. The resulting model-agnostic representations can be fed to either deterministic or generative SR pipelines (see Appendix A.5 for the details of the GeoFAR adaptation to different SR models). As shown in Figure 2a, GeoFAR first applies a frequency-aware projector $P_\psi$ to obtain frequency-aware representations $I_f = P_\psi(I) \in \mathbb{R}^{d \times H \times W}$. Then, GeoFAR encodes geography as implicit neural representations $\mathbf{G} \in \mathbb{R}^{d \times H \times W}$. Finally, we perform pixel-wise frequency-aware conditioning between $I_f$ and $\mathbf{G}$ (*i.e.*, feature-wise modulation) (Perez et al., 2018), getting:

$$M = I_f \odot \mathbf{G}, \tag{1}$$

where $\odot$ denotes per-pixel multiplication. This is followed by three $3 \times 3$ refinement convolutions, yielding $Z$. These representations are then fed into the SR backbone to predict the target image $I_{HR}$, whose learning process is supervised by the Mean Squared Error (MSE) loss by default. In what

follows, we separately explain frequency-aware (Figure 2b) and geography-informed (Figure 2c) representation learning of GeoFAR.

## 3.1 FREQUENCY-AWARE REPRESENTATION LEARNING

Encoding different frequency components separately is beneficial for neural networks to capture patterns at different frequencies (Fuoli et al., 2021; Pan et al., 2022; Patro & Agneeswaran, 2023). For climate SR, we introduce a Frequency-Aware Representation (FAR) learning method that explicitly encodes both high-frequency variability (local-scale climate fluctuations) and low-frequency components (macro climate state) into separate channels, as shown in Figure 2b. Unlike Discrete Wavelet Transforms (DWT), where the decomposed data (LL/LH/HL/HH) is mostly concentrated in the low-frequency subband (LL), we design a Frequency-aware Convolution Kernel (FCK) that achieves multi-granular FAR to avoid low-frequency aggregation in climate data (See Appendix A.8).

The FCK weights are parameterized directly by the bases of the Discrete Cosine Transform (DCT), such that each kernel corresponds to a specific frequency component. By fixing kernel weights with DCT bases of varying frequencies, the convolution acts as a localized filter bank that decouples the input into multiple variants, each one emphasizing a specific frequency band. Such filter bank contains many high-frequency sensitive kernels. By repeatedly sampling high-frequency components in the data, the network is exposed to more high-frequency inputs, which strengthens its reliance on these components for target reconstruction. Formally, for a patch $P \in \mathbb{R}^{h \times w}$ in the 2D grids $I \in \mathbb{R}^{H \times W}$, the 2D DCT is defined as:

$$C(u,v) = c(u)\,c(v) \sum_{x=0}^{N-1} \sum_{y=0}^{N-1} P(x,y)\,\cos\left[\frac{\pi(2x+1)u}{2N}\right] \cos\left[\frac{\pi(2y+1)v}{2N}\right], \qquad (2)$$

where $N$ is the block size, $(x,y)$ are the spatial coordinates, $(u,v)$ denote frequency indices ranging from 0 to $N-1$, with larger values corresponding to higher frequencies. $c(u)$ and $c(v)$ are normalizing constants. The corresponding DCT basis function is given by:

$$B_{u,v}(x,y) = c(u)c(v) \cos\left[\frac{\pi(2x+1)u}{2N}\right] \cos\left[\frac{\pi(2y+1)v}{2N}\right]. \qquad (3)$$

We leverage these bases to define the weights of the convolution kernels: each kernel in the FCK is aligned with one basis function with frequency pair $f_n = (u,v)$. For each patch, the frequency-specific response is obtained via:

$$P_{u,v}(i,j) \;=\; \sum_{x=0}^{N-1} \sum_{y=0}^{N-1} P(x,y)\,B_{u,v}(x,y). \qquad (4)$$

We use FCK to convolve across all patches and apply different frequencies to different channels, yielding frequency-aware representations $I_f \in \mathbb{R}^{d \times H \times W}$ where $d = N^2$.

## 3.2 IMPLICIT NEURAL REPRESENTATION FOR GEOGRAPHY

Implicit neural representation (INR) allows to learn a continuous mapping between the input coordinates and the output signal with neural networks. When trained on sufficient data, INRs can either memorize signals with near-lossless fidelity or learn coordinate-feature functions that generalize across coordinates, powering view synthesis, data compression, and geo-embeddings (Mildenhall et al., 2021; Huang & Hoefler, 2023; Mai et al., 2020; Rußwurm et al., 2024).

Here, we introduce an INR for geography (Geo-INR) that jointly encodes the location- and terrain-driven characteristics of climate processes. To this end, we define climate downscaling on the 3D geographic manifold $\mathcal{M} = \mathbb{S}^2 \times \mathbb{R}$. Each point on $\mathcal{M}$ is $\mathbf{x} = (\lambda, \phi, h)$ with latitude $\lambda \in \left[-\frac{\pi}{2}, \frac{\pi}{2}\right]$ and longitude $\phi \in [-\pi, \pi)$ defined on the unit sphere $\mathbb{S}^2$, while surface elevation $h \in \mathbb{R}$. Our goal is to learn a mapping of each point $\mathbf{x}$ to a geographical representation of dimension $d$:

$$G : \mathcal{M} \to \mathbb{R}^d, \qquad G(\mathbf{x}) \;=\; \text{NN}\big(\text{PE}(\mathbf{x})\big), \qquad (5)$$

where PE is the positional encoding function and NN is a neural network that projects positional encodings into geographical implicit representations. The full map on the 3D geographic manifold

is thus projected to the grid coordinate space as $\mathbf{G} \in \mathbb{R}^{d \times H \times W}$. Concretely, Geo-INR employs fine-scale spherical location encoding and terrain-differential encoding at the target resolution for PE, and their joint use within an NN for geography-informed representations, which are explained separately in the following.

**Spherical location encoding.** We first encode $(\lambda, \phi)$ with a band-limited spherical-harmonic (SH) expansion. Let $\{Y_\ell^m\}_{\ell \geq 0, \, |m| \leq \ell}$ be the real SH basis ($\ell$: degree, $m$: order) orthonormal on $\mathbb{S}^2$:

$$\int_{\mathbb{S}^2} Y_\ell^m(\omega) \, Y_{\ell'}^{m'}(\omega) \, \mathrm{d}\omega = \delta_{\ell\ell'} \delta_{mm'}. \tag{6}$$

We truncate at degree $L$, yielding a multiscale and rotation-aware encoding:

$$\mathbf{Y}_L(\lambda, \phi) \; = \; \left[ Y_\ell^m(\lambda, \phi) \right]_{\ell=0\ldots L, \, m=-\ell\ldots\ell} \in \mathbb{R}^{(L+1)^2}, \tag{7}$$

where smaller $\ell$ captures large-scale patterns on the sphere and larger $\ell$ represents fine details; $Y_\ell^m$ are eigenfunctions of the Laplace-Beltrami operator with eigenvalue $\ell(\ell+1)$ (Atkinson & Han, 2012; Rußwurm et al., 2024). In practice, we use $L = 7$, giving rise to $(L+1)^2 = 64$ SH channels.

**Terrain-differential encoding.** Beyond location, climate states also depend on both absolute elevation and local slope information. In addition to absolute elevation $h$, which is an implicit function of locations $h(\lambda, \phi)$, we add first-order surface derivatives with respect to the latitude and longitude directions to better reflect the slope information:

$$\nabla_{\mathbb{S}^2} h(\lambda, \phi) = \left( \partial_\lambda h, \; \partial_\phi h \right). \tag{8}$$

Elevation and slope information then jointly define the terrain vector:

$$\mathbf{T}(h(\lambda, \phi)) \; = \; \left[ h(\lambda, \phi), \; \partial_\lambda h(\lambda, \phi), \; \partial_\phi h(\lambda, \phi) \right] \in \mathbb{R}^3. \tag{9}$$

A learnable ($3 \times 3$ conv) layer $\Psi : \mathbb{R}^3 \to \mathbb{R}^{(L+1)^2}$ is used to align the elevation with aspect information, producing terrain-differential encoding $\widehat{\mathbf{T}} = \Psi(\mathbf{T})$.

**Geography-informed representation.** We first define the final positional encoding of Geo-INR as the linear fusion of spherical location and terrain-differential encodings:

$$\mathrm{PE}(\mathbf{x}) \; = \; \mathbf{Y}_L(\lambda, \phi) \; + \; \widehat{\mathbf{T}}(h(\lambda, \phi)) \; \in \mathbb{R}^{d_0}, \tag{10}$$

where $d_0$ is unified as the number of SH channels. The positional encodings are then passed through a SIREN (Sitzmann et al., 2020)-based MLP:

$$G(\mathbf{x}) \; = \; f_K \circ f_{K-1} \circ \cdots \circ f_1\big(\mathrm{PE}(\mathbf{x})\big), \quad f_k(\mathbf{z}) = \sin(\omega_k \mathbf{W}_k \mathbf{z} + \mathbf{b}_k), \tag{11}$$

where $\omega_k$ controls the angular period of sinusoidal waves, $\mathbf{W}_k$ and $\mathbf{b}_k$ are learnable weights and bias of the $k^{th}$ layer. Leveraging SIREN's ability to model fine details, we project the positional encodings to the composite geographical representation $G(\mathbf{x}) \in \mathbb{R}^d$ of each point on the 3D manifold that helps the SR backbone reconstruct region-specific structures.

## 4 EXPERIMENTS

### 4.1 EXPERIMENTAL SETUP

**Datasets and experimental settings.** We conduct experiments on three heterogeneous climate databases: **ERA5 Reanalysis** (Hersbach et al., 2020), **PRISM** (PRISM Climate Group, 2025), **CERRA** (Ridal et al., 2024), which are explained in detail in Appendix A.2. We also assess the model's generalization to satellite observations with MODIS (Appendix A.7). To assess the effectiveness of GeoFAR on different *spatial resolutions*, *atmospheric variables*, and *downscaling ratios*, we consider the following settings:

- *Spatial resolutions.* We assess the model on global (ERA5: $5.625°$ to ERA5: $2.8125°$), global-to-local (ERA5: $2.8125°$ to PRISM: $0.75°$), and local (CERRA: 22km to CERRA: 11km) SR scales. For the global and global-to-local settings, we follow the ClimateLearn benchmark (Nguyen et al., 2023): we downscale 2m-temperature (T2m) in ERA5 from $5.625°$ to $2.8125°$ with hourly intervals in the global setting; in the global-to-local setting, we downscale daily max T2m from a reanalysis dataset to an observational dataset (from $2.8125°$ ERA5 data to $0.75°$ PRISM) over the same region at daily intervals. For the local setting, we construct a new high-resolution SR dataset by processing the CERRA database, and use T2m with 3-hour updates.

Table 1: Results on global (ERA5: 5.625° to 2.8125°), global-to-local (ERA5: 2.8125° to PRISM: 0.75°), and local settings (CERRA: 22km to 11km) on T2m (K). We omit Pearson's correlation coefficient ($r$) in global and local downscaling settings since all models achieve $r \approx 1.0$. Values in **bold** and underlined indicate the best and second-best results among learning-based methods. For RMSE and LFD, lower values indicate better performance; for mean bias, values closer to 0.0 are better; and for Pearson's $r$, higher values are better.

| | Global Downscaling | | | Global-to-Local Downscaling | | | | Local Downscaling | | |
|---|---|---|---|---|---|---|---|---|---|---|
| | RMSE | MB | LFD | RMSE | MB | Pearson | LFD | RMSE | MB | LFD |
| Nearest | 3.116 | -0.002 | 11.028 | 2.911 | -0.051 | 0.893 | 9.257 | 0.663 | 0.000 | 11.664 |
| Bilinear | 2.457 | -0.002 | 10.665 | 2.637 | 0.125 | 0.910 | 8.997 | 0.517 | 0.000 | 11.165 |
| ResNet | 1.138 | 0.004 | 9.173 | 1.636 | -0.152 | 0.964 | 8.093 | 0.345 | 0.110 | 10.205 |
| U-Net | 1.103 | 0.004 | 9.114 | 1.501 | -0.094 | 0.970 | 7.953 | 0.272 | 0.068 | 9.769 |
| ViT | 1.121 | 0.009 | 9.154 | 2.163 | -0.147 | 0.937 | 8.652 | 0.380 | 0.033 | 10.496 |
| EDSR | 1.164 | -0.007 | 9.221 | 2.860 | 0.315 | 0.914 | 9.073 | 0.243 | 0.058 | 9.647 |
| FFL | 1.140 | -0.003 | 9.187 | 2.175 | -0.196 | 0.937 | 8.705 | 0.710 | 0.011 | 11.824 |
| SwinIR | 1.117 | 0.002 | 9.141 | 1.879 | -0.087 | 0.952 | 8.392 | 0.212 | -0.001 | 9.450 |
| SRFormer | 1.138 | **0.000** | 9.176 | 1.877 | -0.094 | 0.952 | 8.390 | 0.219 | 0.001 | 9.499 |
| SRGAN | 1.149 | 0.007 | 9.196 | 1.718 | -0.143 | 0.961 | 8.206 | 0.245 | **0.000** | 9.739 |
| DeepSD | 1.396 | 0.002 | 9.590 | 1.955 | -0.198 | 0.949 | 8.451 | 0.344 | -0.004 | 10.401 |
| FACL | 1.373 | 0.156 | 9.490 | 7.240 | 0.133 | 0.928 | 10.182 | 0.700 | -0.053 | 11.761 |
| SmCL | 2.184 | -0.002 | 10.508 | 2.637 | 0.125 | 0.910 | 8.997 | 0.465 | **0.000** | 11.014 |
| STVD | 1.310 | -0.029 | 9.462 | 1.781 | -0.185 | 0.960 | 8.288 | 0.255 | -0.065 | 9.747 |
| ClimateDiffuse | 1.451 | 0.005 | 9.679 | 2.279 | -0.097 | 0.955 | 8.582 | 0.265 | -0.011 | 9.858 |
| DSFNO | 1.265 | 0.019 | 9.397 | 1.546 | **-0.032** | 0.968 | 8.015 | 0.343 | 0.004 | 10.397 |
| GeoFAR[SRGAN] | 1.137 | 0.002 | 9.175 | 1.561 | -0.089 | 0.967 | 7.971 | 0.192 | 0.001 | 9.240 |
| GeoFAR[DSFNO] | 1.126 | -0.003 | 9.160 | 1.474 | -0.121 | 0.971 | 7.904 | 0.190 | 0.000 | 9.234 |
| GeoFAR[ViT] | 1.099 | 0.001 | 9.117 | 1.745 | -0.097 | 0.959 | 8.226 | 0.191 | -0.001 | 9.245 |
| GeoFAR[U-Net] | **1.076** | 0.001 | **9.068** | **1.468** | -0.137 | **0.972** | **7.836** | **0.180** | 0.003 | **9.127** |

- *Atmospheric variables.* In addition to single-variable experiments, we also perform a joint downscaling of multi-variables with CERRA (22km to 11km), including: 2m-temperature (T2m), 10m-u-component-of-wind (10u), 10m-v-component-of-wind (10v), 2m-relative-humidity (Rh2m), surface-presssure (Sp). In complement to the surface-level results, we also perform global-scale downscaling of several pressure-level variables: we downscale geopotential-500hPa (Z500) and temperature-850hPa (T850) in the global setting using ERA5.

- *Downscaling ratios.* In this setting, we explore different downscaling ratios from the first (local) setting on the CERRA dataset. We propose two more challenging downscaling ratios: $4\times$ (from 22km to 5.5km) and $8\times$ (from 44km to 5.5km). This is to evaluate the model's ability to recover high-frequency and high-resolution details from very coarse input.

**Comparison methods.** We compare GeoFAR with both *generic methods* and *climate-oriented methods* for climate SR, leading to one of the most comprehensive comparisons to date in the machine learning community. Generic methods include ResNet (He et al., 2016), U-Net (Ronneberger et al., 2015), ViT (Dosovitskiy et al., 2021), EDSR (Lim et al., 2017), FFL (Jiang et al., 2021), SwinIR (Liang et al., 2021), SRFormer (Zhou et al., 2023), and SRGAN (Ledig et al., 2017). Climate-oriented methods include DeepSD (Vandal et al., 2017), FACL (Yan et al., 2024), and SmCL (Harder et al., 2023). Detailed descriptions of baselines and their implementations can be found in the Appendix A.4.

**Metrics.** We use two types of evaluation metrics: *spatial domain* and *frequency-aware* metrics. By following previous works (Nguyen et al., 2023), Rooted Mean Square Error (RMSE), Mean Bias (MB), and Pearson Coefficient are used to evaluate the SR performance in the spatial domain. Log Frequency Distance (LFD, (Jiang et al., 2021)) and a proposed Wavelet-based Metric are used for the frequency-aware evaluation in Section 4.4.

## 4.2 COMPARATIVE PERFORMANCE

**Results across spatial resolutions.** Table 1 shows the SR results on global (ERA5), global-to-local (ERA5-PRISM), and local (CERRA) scales. GeoFAR is adaptable to both deterministic baselines

Table 2: Results on joint downscaling of multiple variables with CERRA (22km to 11km).

| | T2m (K) | | | 10u (ms$^{-1}$) | | | 10v (ms$^{-1}$) | | | Rh2m (%) | | | Sp (Pa) | | |
| | RMSE | MB | LFD | RMSE | MB | LFD | RMSE | MB | LFD | RMSE | MB | LFD | RMSE | MB | LFD |
|---|---|---|---|---|---|---|---|---|---|---|---|---|---|---|---|
| ViT | 0.457 | 0.033 | 10.966 | 0.341 | 0.002 | 10.395 | 0.355 | -0.011 | 10.472 | 1.799 | -0.012 | 13.708 | 277.719 | -11.007 | 23.808 |
| GeoFAR | 0.262 | 0.001 | 9.859 | 0.184 | 0.000 | 9.163 | 0.186 | 0.000 | 9.187 | 1.215 | -0.003 | 12.921 | 47.922 | 0.375 | 20.291 |

Table 3: Results on ERA5 pressure-level variables (Z500, T850: 5.625° to 2.8125°) and larger downscaling ratios on the CERRA dataset (T2m, K). [S] and [U] denote GeoFAR using SRGAN and U-Net as baselines, respectively.

(a) Results on Z500 and T850 of ERA5

| | Z500 (m$^2$s$^{-2}$) | | | T850 (K) | | |
| | RMSE | MB | LFD | RMSE | MB | LFD |
|---|---|---|---|---|---|---|
| Bilinear | 134.063 | 0.028 | 18.564 | 1.504 | 0.001 | 9.699 |
| U-Net | 49.060 | -0.980 | 16.661 | 0.973 | **0.001** | 8.883 |
| ViT | 51.023 | 0.799 | 16.748 | 0.999 | 0.002 | 8.941 |
| EDSR | 52.602 | -1.758 | 16.794 | 1.040 | -0.015 | 9.010 |
| SRGAN | 50.374 | 0.905 | 16.715 | 1.039 | -0.059 | 9.005 |
| DeepSD | 60.469 | **0.069** | 17.073 | 1.152 | -0.004 | 9.216 |
| GeoFAR[S] | 50.170 | -0.780 | 16.713 | 1.024 | 0.006 | 8.983 |
| GeoFAR[U] | **48.683** | -0.195 | **16.651** | 0.971 | -0.001 | 8.881 |

(b) ×4, ×8 upsampling results on CERRA

| | 22~5.5km (×4) | | | 44~5.5km (×8) | | |
| | RMSE | MB | LFD | RMSE | MB | LFD |
|---|---|---|---|---|---|---|
| Bilinear | 0.610 | 0.000 | 12.878 | 0.861 | 0.000 | 13.578 |
| U-Net | 0.326 | 0.068 | 11.517 | 0.482 | 0.034 | 12.389 |
| ViT | 0.458 | 0.077 | 12.072 | 0.564 | -0.011 | 12.706 |
| EDSR | 0.299 | 0.031 | 11.495 | 0.449 | 0.039 | 12.307 |
| SRGAN | 0.375 | 0.030 | 11.943 | 0.594 | 0.047 | 12.859 |
| DeepSD | 0.388 | 0.024 | 12.024 | 0.579 | 0.034 | 12.815 |
| GeoFAR[S] | 0.253 | -0.002 | 11.172 | 0.434 | **-0.001** | 12.244 |
| GeoFAR[U] | **0.235** | **0.000** | **11.023** | **0.393** | 0.005 | **12.047** |

(*e.g.*, U-Net, ViT, DSFNO) and generative methods (*e.g.*, SRGAN). When applied to simple deterministic baselines such as U-Net, GeoFAR outperforms not only advanced generic SR methods (*e.g.*, SRFormer) but also climate-specific SR methods that incorporate domain priors (*e.g.*, DeepSD with elevation input, SmCL with physics constraints), achieving the state-of-the-art performance across all three settings. These gains are significant, especially in the local-scale fine-grained downscaling (CERRA) setting: the RMSE decreases by 0.092, and the frequency-domain LFD is reduced by 0.642 compared to U-Net. Results also indicate that the gains of geography-informed learning are affected by spatial resolution: as the resolution increases (from ERA5 to CERRA), fine-grained geography-informed representations enable more precise modulation of local-scale SR, yielding larger performance gains.

**Results across atmospheric variables.** In Table 2, we also report results for a joint SR of multiple variables. For a fair comparison, we stack all variables as channels for both the ViT and the ViT-based GeoFAR. Due to fundamentally different physical meanings of variables, jointly super-resolving them with a shared model does not necessarily improve the performance (e.g., the RMSE increases from 0.380 to 0.457 on T2m, ViT). Nevertheless, GeoFAR still provides consistent improvement across variables, especially for variables strongly tied to terrain variability such as surface pressure. In addition to surface-level atmospheric variables, Table 3a provides SR results on different pressure levels (*i.e.*, geopotential-500hPa: Z500, temperature-850hPa: T850). The SR of these pressure-level variables is even more challenging than surface–level variables since their variability is dominated by atmospheric dynamics with comparatively weak high–frequency energy (Nastrom & Gage, 1985; Skamarock et al., 2014), and high sensitivity to location errors since small displacements of ridges or frontal zones can cause large losses (Ebert, 2008). Despite these challenges, our approach (for both UNet and SRGAN adaptation) yields consistent gains over other methods on both Z500 and T850, achieving the state-of-the-art performance.

**Results across downscaling ratios.** We also investigate how models perform on more challenging downscaling ratios. Table 3b shows the SR results on the CERRA high-resolution dataset with ×4, ×8 downscaling ratios. At these ratios, local-scale fine-grained information is heavily smoothed and blurred in the input, and thus pixel-wise errors are nearly doubled as we move from ×2 (Table 1) to ×8. Nevertheless, GeoFAR degrades slowly, and the RMSE is always kept under 0.5, while the Mean Bias is near 0, consistently outperforming the other methods at various scales. This suggests that, even when the input is eight times coarser than the target, GeoFAR still yields locally consistent results relative to real-world atmospheric conditions (with RMSE < 0.5 Kelvin at ×8). Overall, GeoFAR is robust to large downscaling factors, maintaining low error and near-zero bias while consistently outperforming the other methods, which is crucial for real-world applications.

## 4.3 ABLATION STUDY

We quantify the contribution of each GeoFAR component compared to alternative design choices.

**Comparison with alternative designs**. The grey rows in Table 4 show the results for design choices different from ours. As an alternative to our FCK, the Discrete Wavelet Transform (DWT) is commonly used to perform frequency decomposition. However, this strategy (Table 4, w/ DWT) does not improve results on both datasets, probably due to the unbalanced low and high-frequency components in different wavelength channels (LL/LH/HL/HH), aggravating the already skewed frequency distribution. In addition, as an alternative to Geo-INR, we concatenate the input data with the elevation map when super-resolving the target field. Results in Table 4 (w/ Elevation) show that this simple fusion only yields minor gain, mainly due to the domain gap with the target variable and limited interaction with input variables.

**Individual effectiveness.** The black rows in Table 4 show the changes in metrics by adding each component of GeoFAR one by one. We begin with a strong residual baseline (+Residual): instead of regressing the high-resolution image directly, the model predicts the residual between the input and the target, directing capacity to high-frequency corrections. Based on this, incorporating frequency-aware (+ FCK) and geography-informed (+ Geo-INR) representations further reduces the RMSE on both global and local datasets. Jointly modeling location and elevation in INR performs better than the

Table 4: Ablation study on both local-scale and global-scale T2m (K) downscaling ($\times 2$).

| | CERRA | | | ERA5 | | |
|---|---|---|---|---|---|---|
| | RMSE | MB | LFD | RMSE | MB | LFD |
| ViT | 0.380 | 0.033 | 10.496 | 1.125 | **0.001** | 9.184 |
| w/ DWT | 0.434 | 0.013 | 10.787 | 1.139 | 0.007 | 9.186 |
| w/ Elevation | 0.381 | 0.018 | 10.451 | 1.117 | -0.005 | 9.146 |
| + Residual | 0.233 | 0.003 | 9.664 | 1.110 | 0.003 | 9.141 |
| + FCK | 0.216 | **0.000** | 9.493 | 1.100 | **0.001** | 9.118 |
| + 2D-INR | 0.198 | -0.001 | 9.310 | **1.099** | 0.008 | 9.118 |
| + Geo-INR | **0.191** | -0.001 | **9.245** | **1.099** | 0.001 | **9.113** |

location-only method (+ 2D-INR). Fine-grained local-scale SR benefits more from our approach by learning a more accurate mapping between the geographical data and target variables.

## 4.4 BEYOND GENERIC METRICS: IN-DEPTH ANALYSES OF GEOFAR EFFECTIVENESS

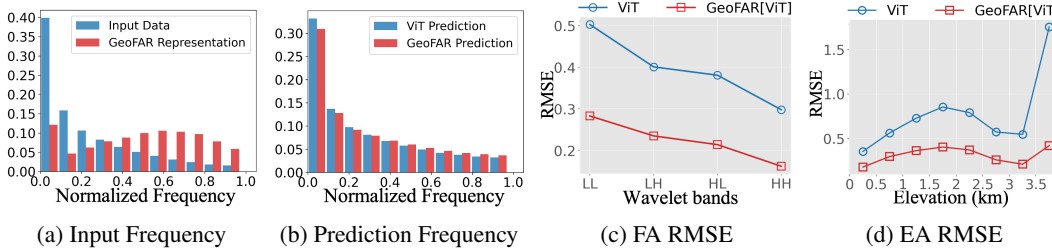

(a) Input Frequency    (b) Prediction Frequency    (c) FA RMSE    (d) EA RMSE

Figure 3: **Frequency and elevation encoding analyses.** On CERRA, we compare the frequency distribution of (a) input data with GeoFAR representations; and (b) the output predictions of Geo-FAR[ViT] and ViT. We also perform: (c) frequency-aware (FA) evaluation on Wavelet decomposed subbands; and (d) elevation-aware (EA) evaluation across increasing elevations by comparing Geo-FAR[ViT] with ViT.

To further investigate the effectiveness of GeoFAR, we conduct in-depth frequency-aware and geography-aware analyses, beyond generic SR metrics. We first perform a frequency analysis on CERRA to assess how GeoFAR affects learning in the frequency domain. In climate data, low-frequency components are usually related to large-scale structures (e.g., meridional temperature gradients), whereas the high-frequency components capture localized events such as sharp frontal zones and topographically induced anomalies. Figure 3a shows the frequency distributions of the raw input data and the frequency-aware climate representations: the high-frequency information of the input data has been enhanced by learning frequency-aware variants of the input data, mitigating the low-frequency bias in DNNs. This mitigation is also reflected in the output predictions as shown in Figure 3b: GeoFAR generates more high-frequency components than ViT. We also perform a frequency-aware evaluation to analyze how accurate the enhanced high-frequency infor-

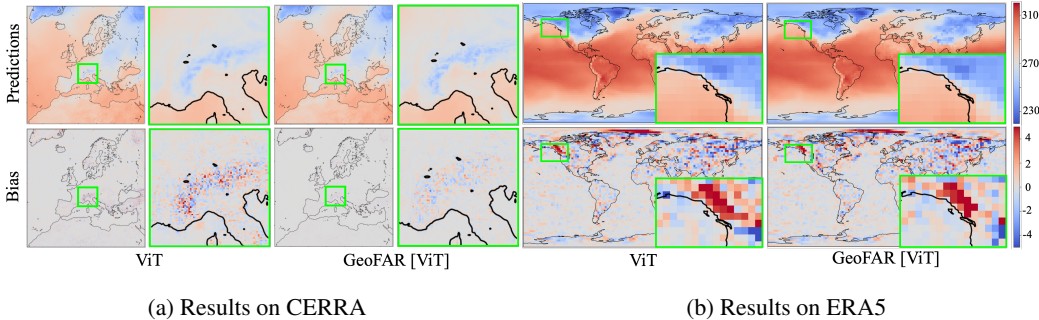

(a) Results on CERRA  (b) Results on ERA5

Figure 4: **An example of super-resolved climate data.** Compared to ViT, GeoFAR addresses the oversmoothing by recovering high-frequency details in complex terrains on (a) CERRA and (b) ERA5. Meanwhile, GeoFAR significantly mitigates the prediction bias in both datasets.

mation is: we first decouple different frequency components in predictions with DWT into four bands (LL/LH/HL/HH), then calculate RMSE for each band. Results in Figure 3c show a consistent reduction of RMSE across frequency bands, with high-frequency subband (HH) showing the largest relative improvement. This indicates that GeoFAR improves the reconstruction of both large-scale and fine-scale local climate structures.

Second, we perform analyses to investigate how GeoFAR performs in different regions. In Figure 3d, we first group regions according to elevation ranges, then evaluate the RMSE within each group: the RMSE has been consistently reduced from lowlands to high plateaus, while the most significant gains are observed at high elevations: in regions over 3km elevation, RMSE significantly drops from 1.755 to 0.210, showing the particular benefit of terrain-guided Geo-INR in complex mountainous ar-

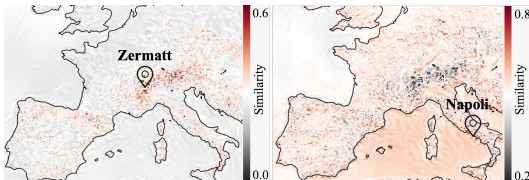

Figure 5: **GeoFAR representation similarities.** Red (black) indicates higher (lower) similarity, for Zermatt (left) and Napoli (right).

eas. In Europe, such high-elevation areas are concentrated in the Alps and Pyrenees, where rugged topography induces rapidly varying atmospheric fields, making SR particularly challenging. The super-resolved climate data in Figure 4a shows that even in the challenging Alps, GeoFAR reproduces fine-grained temperature patterns between adjacent valleys and ridges and recovers local details under geographic conditioning. To further probe the learned geography-informed representations, we measure the dot-product similarities of a predefined location with respect to the full map. Figure 5 shows high similarity between a city in the Alps (Zermatt) and other mountainous regions (*e.g.*, Pyrenees); in comparison, the representation of Napoli (a relatively flat coastal city) aligns more with other flat areas and regions close to the Mediterranean Sea.

## 5  CONCLUSION AND OUTLOOK

Super-resolving climate data is crucial for fine-grained mapping of local climate states. In this paper, we introduce GeoFAR, a model-agnostic approach that learns frequency-aware and geography-informed representations for high-fidelity climate super-resolution. Across three datasets spanning multiple spatial resolutions, atmospheric variables, and downscaling factors, GeoFAR consistently improves both deterministic and generative baselines by recovering high-frequency structure and modulating outputs specific to real-world location and terrain variations. With its broad adaptability, GeoFAR lays the groundwork for precise, geography-aware, and physically consistent climate downscaling. We hope our findings can provide valuable insights for preserving the inherent relationships between geographic factors and climate states during downscaling, and inform broader applications such as weather forecasting and climate projections.

## 6 ACKNOWLEDGEMENT

We thank Hugo Porta for the discussion and the code for downloading the MODIS dataset. We thank all the anonymous reviewers for their constructive comments. We acknowledge the support from CSC and the funding under the Horizon Europe grant 101213369 DVPS.

### REPRODUCIBILITY STATEMENT

We provide all necessary details to ensure reproducibility. Model descriptions and experimental setups are described in the main text and the appendix. The CERRA dataset and code are available at `https://eceo-epfl.github.io/GeoFAR/`.

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

## A  APPENDIX

The appendix contains the following information:

- A summary of notations and variables (Section A.1)

- Data (Section A.2)

- Evaluation metrics (Section A.3)

- Comparison methods (Section A.4)

- Detailed method description (Section A.5)

- Implementation details (Section A.6)

- Additional experiments: robustness and generalization, hyper-parameter analysis, more ablations for Geo-INR, computational cost, physical fidelity (Section A.7)

- Additional visualization: super-resolution results from both deterministic and generative models, frequency-aware representations, terrain information (Section A.8)

- Limitations and future works (Section A.9)

- LLM usage (Section A.10)

### A.1  A SUMMARY OF NOTATIONS AND VARIABLES

We summarize the main notations in the paper in Table 5, and summarize the meteorological variables studied in this work in Table 6.

Table 5: Summary of main notations.

| Symbol | Description | Dimension |
|---|---|---|
| $I$ | Low-resolution input field | $\mathbb{R}^{H \times W}$ |
| $I_{\text{HR}}$ | High-resolution target field | $\mathbb{R}^{H' \times W'}$ |
| $I_f$ | Frequency-aware representation | $\mathbb{R}^{d \times H \times W}$ |
| $P_\psi$ | Frequency-aware projection | $P_\psi : I \mapsto I_f$ |
| $\mathbf{G}$ | Geographical implicit neural representation | $\mathbb{R}^{d \times H \times W}$ |
| $M$ | Frequency-aware geography-informed representation | $\mathbb{R}^{d \times H \times W}$ |
| $P$ | Data patch | $\mathbb{R}^{h \times w}$ |
| $C$ | Discrete cosine transformed data patch | $\mathbb{R}^{N \times N}$ |
| $N$ | DCT block size | scalar |
| $(x, y)$ | Spatial coordinates | $\mathbb{R}^2$ |
| $(u, v)$ | Frequency indices | $\mathbb{R}^2$ |
| $B_{u,v}$ | DCT basis | $\mathbb{R}^{N \times N}$ |
| $P_{u,v}$ | Frequency-specific response | $\mathbb{R}^{N \times N}$ |
| $(\lambda, \phi, h)$ | Latitude, longitude, and elevation coordinate | $\mathbb{R}^3$ |
| $\mathcal{M}$ | 3D geographic manifold | $\mathbb{S}^2 \times \mathbb{R}$ |
| $G$ | Implicit neural mapping | $\mathcal{M} \to \mathbb{R}^d$ |
| PE | Positional encoding | $\text{PE} : \mathbb{R}^3 \to \mathbb{R}^{d_0}$ |
| NN | Neural network | $\text{NN} : \mathbb{R}^{d_0} \to \mathbb{R}^d$ |
| $Y_\ell^m$ | Spherical harmonic basis | $\mathbb{S}^2$ |
| $\mathbf{Y}_L$ | Truncated spherical harmonics encoding | $\mathbb{R}^{(L+1)^2}$ |
| $\mathbf{T}$ | Terrain vector | $\mathbb{R}^3$ |
| $\widehat{\mathbf{T}}$ | Terrain-differential encoding | $\mathbb{R}^3$ |
| $\Psi$ | Learnable layer for terrain-differential encoding | $\Psi : \mathbb{R}^3 \to \mathbb{R}^{(L+1)^2}$ |
| $f_k$ | Sinusoidal activation function | $f_k : \mathbb{R}^d \to \mathbb{R}^d$ |

Table 6: Summary of meteorological variables.

| Abbreviation | Full Name | Source | Unit |
|---|---|---|---|
| T2m | 2m temperature | ERA5, CERRA | K |
| Z500 | geopotential 500hPa | ERA5 | $m^2s^{-2}$ |
| T850 | temperature 850hPa | ERA5 | K |
| 10u | 10m u component of wind | CERRA | $ms^{-1}$ |
| 10v | 10m v component of wind | CERRA | $ms^{-1}$ |
| Rh2m | 2m relative humidity | CERRA | % |
| Sp | surface pressure | CERRA | Pa |
| LST | land surface temperature | MODIS | K |
| SKT | skin temperature | ERA5-Land | K |

## A.2 DATA

### A.2.1 DATASET DETAILS

**ERA5 Reanalysis.** ERA5 Reanalysis (Hersbach et al., 2020) is maintained by the European Center for Medium-Range Weather Forecasting (ECMWF). ERA5 is a reanalysis that combines diverse observational data with forecasts from the state-of-the-art Integrated Forecasting System (IFS). It provides the estimation of the state of the atmosphere and land-surface variables at any given time. In its raw format, ERA5 provides hourly data from 1979 to the present on a $0.25°$ global grid, covering a range of atmospheric variables at 37 pressure levels in addition to surface conditions. We preprocess the 2m-temperature (T2m), geopotential at 500hPa (Z500), and temperature at 850hPa (T850) in ERA5 according to the ClimateLearn benchmark into $5.625°$ ($32\times64$) and $2.8125°$ ($64\times128$) global grids with hourly updates for climate downscaling, we use the years 1981-2015 for training, the year 2016 for validation, and 2017-2018 for testing.

**PRISM.** PRISM (PRISM Climate Group, 2025) is a dataset of observed atmospheric variables, including precipitation and temperature, covering the conterminous United States at varying spatial and temporal resolutions from 1895 to the present. At its highest publicly available resolution, PRISM provides daily data on a $4 \times 4$ km grid, corresponding to a matrix of $621 \times 1405$ cells. Following the ClimateLearn benchmark, we process daily max T2m in PRISM to $0.75°$. We crop the corresponding regions in ERA5 to align with PRISM when downscaling. The training period is from 1981 to 2015, validation is in 2016, and the testing period is from 2017 to 2018.

**CERRA.** Copernicus European Regional Reanalysis for Europe (CERRA) is a pan-European regional reanalysis produced by C3S/ECMWF at 5.5 km horizontal resolution, with the core production spanning September 1984 to June 2021. For single-variable experiments (Table 1 and Table 3b), we specifically employ the 2-m air temperature (t2m) field available as a single-level variable via the C3S Climate Data Store. For our experiments, we use bilinear interpolation to process the native 5.5 km data ($1069 \times 1069$) to grids of 5.5 ($1068 \times 1068$), 11 ($534 \times 534$), 22 ($267 \times 267$), and 44 ($133 \times 133$) km with 3-hour updates to define multiple evaluation settings, we use 2010-2017 for training, 2018-2021 for validation and testing. For multi-variable experiments (Table 2), we employ the 2m-temperature (T2m), 10m-u-component-of-wind (10u), 10m-v-component-of-wind (10v), 2m-relative-humidity (Rh2m), and surface-presssure (Sp). The native 5.5 km data are further preprocessed with bilinear interpolation to grids of 11 ($534 \times 534$), 22 ($267 \times 267$) km with 3-hour updates for the $2\times$ super-resolution. The training set is from 2010 to 2017, and the val/test sets are from 2018 to 2021, leading to a large-scale multi-variable downscaling dataset of roughly 200GB.

**MODIS.** Moderate Resolution Imaging Spectroradiometer (MODIS) is a satellite from NASA, whose data are used to create a number of operational products. To assess the model's generalization ability on this data source, we construct a new dataset aiming at downscaling the daily skin temperature (SKT) from ERA5-Land to the MODIS land surface temperature (LST). Specifically, we download both datasets over Switzerland, which is a very challenging region with complex atmospheric processes, due to its complex alpine topography. We preprocess ERA5-Land to 11 km resolution ($21\times46$), and MODIS to roughly 1.1 km resolution ($210\times460$) via bilinear interpolation, leading to a challenging setting of $\times10$ downscaling. The training set is from 2018 to 2022, and the testing set is from 2023.

## A.2.2 DATA PRE-PROCESSING

We normalize both the input and target image $I \in \mathbb{R}^{H \times W}$ to stabilize model training:

$$\tilde{I}(x,y) = \frac{I(x,y) - \mu}{\sigma + \varepsilon}, \qquad (x,y) \in \{1, \ldots, H\} \times \{1, \ldots, W\}, \tag{12}$$

where $\mu$ and $\sigma$ are the mean and standard deviation (computed over the training set), and $\varepsilon > 0$ ensures numerical stability. In addition, we add a constant offset (1.0) to the frequency-aware representation $I_f$, to prevent near-zero entries in $I_f$ from nullifying the contribution of the geographical embedding during the per-pixel product.

## A.3 EVALUATION METHODS

In the introduction, we use the Laplace operator to filter the prediction bias in Figure 1. In the experiment section, we use two types of evaluation metrics: *spatial domain* and *frequency-aware* metrics.

### A.3.1 LAPLACE OPERATOR

In Figure 1, we use the Laplace operator as a high-pass filter to visualize the prediction bias. The Laplace operator is written as:

$$\nabla^2 f = \frac{\partial^2 f}{\partial x^2} + \frac{\partial^2 f}{\partial y^2}. \tag{13}$$

- $f$ indexes the prediction bias in 2D;
- $\frac{\partial^2 f}{\partial x^2}$ is the second derivative of $f$ with respect to $x$-direction measuring curvature or variation along $x$.
- $\frac{\partial^2 f}{\partial y^2}$ is the second derivative along the $y$-direction.

The Laplacian highlights details, high-frequency features, and regions where prediction bias changes sharply, while smoothing out low-frequency components.

### A.3.2 SPATIAL DOMAIN METRICS

- **RMSE (Rooted Mean Square Error)**:

$$\text{RMSE} = \frac{1}{N} \sum_{k=1}^{N} \sqrt{\frac{1}{H \times W} \sum_{i=1}^{H} \sum_{j=1}^{W} L(i) \left( \tilde{X}_{k,i,j} - X_{k,i,j} \right)^2}, \tag{14}$$

  - $k = 1, \ldots, N$ indexes samples;
  - $i = 1, \ldots, H$ and $j = 1, \ldots, W$ index grid rows and columns;
  - $X_{k,i,j}$ is the ground-truth value at pixel $(i,j)$ of sample $k$;
  - $\tilde{X}_{k,i,j}$ is the corresponding prediction;
  - $L(i)$ is an optional latitude (row) weight. We use the normal RMSE without reweighting, so $L(i)$ is set to 1.

- **MB (Mean Bias)**: measures the difference between the spatial mean of the prediction and the spatial mean of the ground truth. A positive mean bias shows an overestimation, while a negative mean bias shows an underestimation of the mean value.

$$\text{MB} = \frac{1}{N \times H \times W} \sum_{k=1}^{N} \sum_{i=1}^{H} \sum_{j=1}^{W} \tilde{X}_{k,i,j} - \frac{1}{N \times H \times W} \sum_{k=1}^{N} \sum_{i=1}^{H} \sum_{j=1}^{W} X_{k,i,j}, \tag{15}$$

  where $N$ is the number of samples, $H \times W$ the grid size, $X_{k,i,j}$ the ground truth, and $\tilde{X}_{k,i,j}$ the prediction.

- **Pearson Coefficient**: measures the correlation between the prediction and the ground truth. We first flatten the prediction $\tilde{X}$ and ground truth $X$, and compute the metric as follows:

$$\rho_{\tilde{X},X} = \frac{\text{cov}(\tilde{X}, X)}{\sigma_{\tilde{X}} \sigma_X},\qquad(16)$$

where $\text{cov}(\cdot, \cdot)$ is the covariance and $\sigma_{\tilde{X}}$, $\sigma_X$ are standard deviations. The coefficient measures linear association, with range $[-1, 1]$ (1 = perfect positive, 0 = no linear correlation, $-1$ = perfect negative).

### A.3.3 FREQUENCY-AWARE METRICS

- **Log Frequency Distance (LFD)** (Jiang et al., 2021):

$$\text{LFD} = \log \left[ \frac{1}{H \times W} \left( \sum_{u=0}^{H-1} \sum_{v=0}^{W-1} |F_p(u, v) - F_g(u, v)|^2 \right) + 1 \right],\qquad(17)$$

where $F_p(u, v)$, $F_g(u, v)$ denote the Fourier transformed predictions and ground truth, respectively, the logarithm is only used to scale the value into a reasonable range. A lower LFD is better.

- **Wavelet-based Metric**: In order to evaluate the performance on different frequency bands in Figure 3c, we use the Discrete Wavelet Transform (DWT) to decouple the predicted image and the ground truth target into four bands (LL/LH/HL/HH), then evaluate the RMSE for each band.

- **Kinetic Energy Spectral RMSE (RMSE$_{\text{KE}}$)**: To assess the scale-dependent consistency of the predicted wind fields, we compare their isotropic kinetic energy (KE) spectra (Bolgiani et al., 2022) to the one of the ground truth. Given horizontal winds $u(x, y)$ and $v(x, y)$, we remove the spatial mean, apply a 2D Hann window, and compute the 2D Fourier transform to obtain the KE power spectral density $\text{PSD}(k_x, k_y) = \frac{1}{2} \big( |\hat{u}(k_x, k_y)|^2 + |\hat{v}(k_x, k_y)|^2 \big)$. Using the radial wavenumber $k_r = \sqrt{k_x^2 + k_y^2}$, we perform ring averaging in Fourier space to obtain an isotropic KE spectrum $E_b(k_i)$ for batch index $b$ and wavenumber bin $i$. We then define the kinetic energy spectral RMSE between predicted $E_b^{\text{p}}(k_i)$ and target spectra $E_b^{\text{t}}(k_i)$ as

$$\text{RMSE}_{\text{KE}} = \sqrt{\frac{1}{B\,N_k} \sum_{b=1}^{B} \sum_{i=1}^{N_k} \big( E_b^{\text{p}}(k_i) - E_b^{\text{t}}(k_i) \big)^2}.\qquad(18)$$

This error directly measures how well the model reproduces the distribution of kinetic energy across spatial scales.

## A.4 COMPARISON METHODS

We re-implemented both *generic methods* and *climate-oriented methods* for climate SR, making it one of the most comprehensive comparisons to date in the machine learning community.

### A.4.1 GENERIC METHODS

Generic methods include ResNet (He et al., 2016), U-Net (Ronneberger et al., 2015), ViT (Dosovitskiy et al., 2021), EDSR (Lim et al., 2017), FFL (Jiang et al., 2021), SwinIR (Liang et al., 2021), SRFormer (Zhou et al., 2023), and SRGAN (Ledig et al., 2017). The implementations of ResNet, U-Net, and ViT follow the ClimateLearn paper (Nguyen et al., 2023) to ensure a fair comparison.

**EDSR** EDSR (Lim et al., 2017) is a refined version of the ResNet architecture for SR. EDSR removes batch normalization layers and simplifies residual blocks, making it better suited for single-image super-resolution. The simplified architecture enables the construction of deeper networks stabilized via residual scaling, which significantly boosts SR results. We implement EDSR with 28 residual blocks which aligns with the implementation of ResNet in ClimateLearn.

**FFL**  Focal Frequency Loss (FFL (Jiang et al., 2021)) encourages models to adaptively focus on frequency components that are difficult to synthesize. By down-weighting the low-frequency components and emphasizing the higher-frequency components, FFL helps preserve and recover fine textures and details in reconstructed images. Though primarily applied to generative models (*e.g.*, VAE, pix2pix, StyleGAN2). We build the FFL based on the ViT model in ClimateLearn.

**SwinIR**  SwinIR (Liang et al., 2021) is based on the Swin Transformer (Liu et al., 2021). The architecture comprises three stages: shallow feature extraction, deep feature extraction using residual Swin Transformer blocks (RSTBs), and high-quality image reconstruction. SwinIR leverages the Swin Transformer's hierarchical shifted-window attention within a residual learning framework. We set the depths to [3, 3, 3, 3] and num_heads to 4 for better alignment with the parameters of ViT in ClimateLearn.

**SRFormer**  SRFormer (Zhou et al., 2023) introduces a permuted self-attention (PSA), which strikes an appropriate balance between the channel and spatial information for self-attention. PSA enables efficient computation of long-range pairwise correlations within significantly larger attention windows, achieving better coverage of spatial context. We set the depths to [3, 3, 3, 3] and num_heads to 4 for better alignment with the parameters of ViT in ClimateLearn.

**SRGAN**  SRGAN (Ledig et al., 2017) is the first generative adversarial network (GAN) framework designed to produce photo-realistic high-resolution images at $4\times$ upscaling factors. SRGAN employs a perceptual loss that combines a content loss based on high-level feature maps from a pretrained VGG network and an adversarial loss from a discriminator, encouraging reconstructions that lie on the manifold of natural images. To balance the model parameters, we build the SRGAN with 16 residual blocks in the generator and 8 convolutional blocks in the discriminator.

### A.4.2  CLIMATE-ORIENTED METHODS

Climate-oriented methods include DeepSD (Vandal et al., 2017), FACL (Yan et al., 2024), and SmCL (Harder et al., 2023).

**DeepSD**  DeepSD (Vandal et al., 2017) adapts stacked Super-Resolution Convolutional Neural Networks (SRCNNs) for climate data downscaling. Each SRCNN performs a small upscaling step, and stacking them enables large-scale resolution enhancement, similar to multi-stage image super-resolution pipelines. The method also integrates high-resolution static features such as elevation as auxiliary inputs, analogous to conditioning strategies in SR models. We train the DeepSD with the original architecture proposed in  (Vandal et al., 2017).

**FACL**  Fourier Amplitude and Correlation Loss (FACL (Yan et al., 2024)), comprises two complementary components: Fourier Amplitude Loss (FAL) and Fourier Correlation Loss (FCL). FAL constrains the Fourier amplitude of model outputs to better capture spectral content, while FCL addresses missing or misaligned phase information by enforcing correlation structures in the frequency domain. We build the FACL based on the ViT model in ClimateLearn.

**SmCL**  SmCL (Harder et al., 2023) introduces a climate downscaling method that incorporates hard physical constraints (*e.g.*, conservation of mass or energy) directly into the model architecture, ensuring physically consistent high-resolution outputs. We build the SmCL based on the ViT model in ClimateLearn.

**ClimateDiffuse**  ClimateDiffuse (Watt & Mansfield, 2024) performs climate downscaling as a generative task with a diffusion model, providing both target prediction and probability estimation, which can be used for risk assessment. ClimateDiffuse serves as a simple yet effective baseline to assess the capability of diffusion-based models for climate variables.

**STVD**  STVD (Srivastava et al., 2024) extends recent video diffusion methods to super-resolving precipitation data by combining a deterministic model with a temporally-conditioned diffusion model to generate high-frequency patterns. We adapt this model to perform temporally independent climate downscaling that aligns with the setting of this paper.

**DSFNO**  DSFNO (Yang et al., 2024) formulates statistical downscaling as learning a resolution-agnostic operator in Fourier space. It is trained on low-/high-resolution pairs with a small, fixed upsampling factor and is then applied in a zero-shot fashion to downscale inputs to unseen higher resolutions. In this paper, we train DSFNO on our datasets and use it as the backbone on which we build our proposed method GeoFAR.

## A.5  DETAILED METHOD DESCRIPTION

In this section, we explain how to plug our approach into deterministic and generative baselines given the geography-informed representation $Z$:

**Deterministic downscaling** learns a single mapping by empirical risk minimization:

$$\hat{\theta} \;=\; \arg\min_{\theta} \; \mathbb{E}_{(I,\, I_{\mathrm{HR}})} \big[\; \mathcal{L}\big(f_{\theta}(Z),\, I_{\mathrm{HR}}\big) \;\big], \tag{19}$$

where $\mathcal{L}$ is typically an $\ell_n$-based loss, and $f_{\theta}$ upsamples the input to the HR grid. Geo-INR steers $f_{\theta}$ toward geography-consistent high frequency details while maintaining fidelity.

**Generative downscaling** like SRGAN (Ledig et al., 2017) can also be plugged with our method for geo-conditioned learning to better generate plausible high-frequency details. Let $g_{\theta}$ be the Geo–conditioned generator that maps $Z$ to an HR image, and $D_{\phi}$ a discriminator. The generator update minimizes fidelity loss via the Mean Squared Error (MSE):

$$\min_{\theta} \; \mathcal{L}_{\mathrm{G}} \;=\; \mathbb{E}\big[\, \| \, g_{\theta}(Z) - I_{\mathrm{HR}} \, \|_2^2 \,\big], \tag{20}$$

while the discriminator update minimizes the binary cross-entropy (BCE) loss corresponding to the classic minimax game:

$$\min_{\phi} \; \mathcal{L}_{\mathrm{D}} \;=\; -\mathbb{E}\big[\, \log D_{\phi}(I_{\mathrm{HR}}) \,\big] - \mathbb{E}\big[\, \log \big(1 - D_{\phi}(g_{\theta}(Z))\big) \,\big].$$

This updates the adversarial training signal on $D_{\phi}$ (as in SRGAN) while directing $g_{\theta}$ strictly toward high-fidelity reconstruction via MSE. The Geo-INR modulation anchors the generation of high-resolution data to geography information (location and terrain), helping recover region-specific high-frequency details and mitigating GAN artifacts.

## A.6  IMPLEMENTATION DETAILS

All experiments are run on a working station with one NVIDIA RTX A5500 GPU and an Intel Xeon Silver 4410Y CPU (1 socket, 12 physical cores, 2 threads/core).

We implement our method and re-implement baselines using the ClimateLearn training pipeline (Nguyen et al., 2023). For **ERA5** and **PRISM**, we train for 50 epochs with batch size 16, learning rate $2 \times 10^{-4}$, and weight decay $1 \times 10^{-4}$. For **CERRA** ($\times 2$), we train for 20 epochs with batch size 4, learning rate $2 \times 10^{-4}$, and weight decay $2 \times 10^{-4}$. For **CERRA** ($\times 4$, $\times 8$), we keep the learning rate and weight decay at $2 \times 10^{-4}$ and reduce the batch size to 1 due to memory constraints. For **ERA5 to MODIS**, we train for 50 epochs with batch size 8, learning rate $2 \times 10^{-4}$, and weight decay $1 \times 10^{-4}$. Early stopping is applied in all settings if the validation loss does not decrease for 5 consecutive epochs.

In GeoFAR, we set the embedding dimension $d$ to 64, and use two layers of SIREN to embed the geographical information. For FCK, we set the kernel size ($N$) to 8, the stride to 1 and the zero-padding to length 4, so that we generate 64 ($N^2$) channels aligned with the dimension $d$ of Geo-INR.

## A.7  ADDITIONAL EXPERIMENTS

**Robustness and generalization.**  We investigate GeoFAR's robustness to random initialization, its sensitivity to smoothed low-resolution inputs, and its ability to generalize to remote sensing observations:

- Table 7a reports the performance of GeoFAR on downscaling CERRA T2m from 22 km to 11 km under different initialization seeds. The performance is stable across the different seeds: RMSE remains unchanged, the mean bias and LFD only fluctuate within $\pm 0.002$.

Table 7: Robustness and generalization analysis that investigates GeoFAR's robustness to (a) initialization seeds and (b) smoothed inputs (Methods with * denote results with original inputs), as well as (c) models' generalization on downscaling ERA5 (SKT, 11km) to MODIS land surface temperature (LST, 1.1km).

| (a) Initialization seeds | | | | (b) Smoothed inputs (CERRA ×2) | | | | | (c) ERA5 to MODIS | | | |
|---|---|---|---|---|---|---|---|---|---|---|---|---|
| Seed | RMSE | MB | LFD | | Method | RMSE | MB | LFD | | Method | RMSE | Pearson | MB |
| 0 | 0.191 | -0.001 | 9.245 | | ViT* | 0.380 | 0.033 | 10.496 | | ViT | 3.474 | 0.771 | 1.822 |
| 23 | 0.191 | 0.000 | 9.243 | | ViT | 0.436 | 0.071 | 10.824 | | U-Net | 3.715 | 0.720 | 1.696 |
| 149 | 0.191 | 0.001 | 9.242 | | GeoFAR[ViT]* | 0.191 | -0.000 | 9.245 | | GeoFAR[ViT] | 3.266 | 0.766 | 0.309 |
| 71 | 0.191 | -0.002 | 9.244 | | GeoFAR[ViT] | 0.213 | 0.003 | 9.466 | | GeoFAR[U-Net] | 2.718 | 0.844 | 0.717 |

Table 8: Additional experiments on CERRA (×2) that investigate (a) the effects of the number of embeddings on accuracy and inference speed (FPS), (b) the effects of different positional encodings, and (c) the utility of information from terrain vectors in GeoFAR. 'Elevation' means only using location and elevation as the input of INR, 'Vectors' mean using the location and terrain-differential vector as input.

| (a) Embedding dimension | | | | | (b) Positional encoding | | | | | (c) Terrain vector | | | |
|---|---|---|---|---|---|---|---|---|---|---|---|---|---|
| Dim. | RMSE | MB | LFD | FPS | PE | RMSE | MB | LFD | | Vector | RMSE | MB | LFD |
| 36 | 0.192 | -0.001 | 9.254 | 11.3 | Direct | 0.195 | -0.001 | 9.256 | | w/o | 0.198 | -0.001 | 9.310 |
| 64 | 0.191 | -0.001 | 9.245 | 11.1 | Space2Vec | 0.192 | 0.001 | 9.253 | | Elevation | 0.193 | -0.001 | 9.260 |
| 100 | 0.190 | 0.000 | 9.235 | 10.7 | SH | 0.191 | -0.001 | 9.245 | | Vectors | 0.191 | -0.001 | 9.245 |

- Table 7b compares performances when using smoothed inputs versus the original inputs (CERRA 22 km) for downscaling to CERRA 11 km. In the main paper, we constructed the CERRA dataset by downsampling the raw data from 5.5km to resolution variants for SR using bilinear interpolation. Here, we assess robustness to smoothed input simulations by first downsampling the raw data to 11 km and then to 22 km, applying Gaussian smoothing with $\sigma = 1$ at each stage to obtain the smoothed inputs. Compared to the results without Gaussian smoothing, both methods suffer from a slight decrease of performance (RMSE, ViT: $0.380 \rightarrow 0.436$, GeoFAR: $0.191 \rightarrow 0.213$). However, GeoFAR still substantially improves the baseline (RMSE: $0.436 \rightarrow 0.213$) and shows better robustness to input degradation (RMSE: ViT 0.380 vs. 0.436, GeoFAR: 0.191 vs. 0.213).

- To analyze the generalization ability on diverse data products, Table 7c presents the results on downscaling from ERA5-Land (SKT, 11km) to MODIS (LST, 1.1km), i.e. when training the model to downscale from reanalysis products (ERA5) to remote sensing observations (MODIS). This task is more challenging than other settings due to large domain and resolution gaps between the input and output, leading to higher errors than observed on the ERA5 and CERRA datasets (Table 1). Nonetheless, this challenging dataset still benefits from GeoFAR, with RMSE and Mean Bias markedly reduced with respect to both the ViT (RMSE: $3.474 \rightarrow 3.266$) and U-Net (RMSE: $3.715 \rightarrow 2.718$) baselines.

**Hyper-parameter analysis.** Table 8a presents the effects of the embedding dimension $d$ in Equation 5. We set $L \in \{5, 6, 7\}$, yielding $(L+1)^2 = \{36, 64, 100\}$ spherical-harmonics (SH) channels (i.e., $d$). Increasing SH channels from 36 to 100 reduces RMSE by 0.002 and LFD by 0.019, while slightly decreasing the inference speed by 0.6 frames per second (FPS). To balance accuracy and cost, we use $d = 64$ in all experiments.

**More ablations for Geo-INR.** Table 8b compares three location encoders (latitude/longitude based): *Direct* stacks $(\text{lat}, \text{lon})$ into $d$ channels as INR input; *Space2Vec* uses sine and cosine functions of different frequencies to encode positions in space (Mai et al., 2020); *SH* uses spherical harmonics (Rußwurm et al., 2024). Among them, *SH* achieves the best overall performance, so we adopt it as our default location encoder. Table 8c evaluates terrain-aware embeddings on top of the SH location encoder. Removing terrain-differential cues degrades RMSE by $\sim 4\%$ (from 0.191 to 0.198) and worsens the frequency metric LFD (from 9.245 to 9.310). Using only elevation partially recovers accuracy (RMSE 0.193). The composite terrain-differential embedding yields the best results (RMSE 0.191, LFD 9.245).

Table 9: A comparison of model size (millions of parameters) and inference speed (frames per second, FPS). All models are tested on a working station with RTX A5500 GPU, on the CERRA dataset with input image size $534 \times 534$.

| Model | ResNet | U-Net | ViT | SRGAN | EDSR | SwinIR | GeoFAR[U-Net] | GeoFAR[ViT] | GeoFAR[SRGAN] |
|---|---|---|---|---|---|---|---|---|---|
| #Params (M) | 8.3 | 11.6 | 2.6 | 6.1 | 9.0 | 2.3 | 12.5 | 3.5 | 7.0 |
| Speed (FPS) | 3.8 | 14.1 | 15.0 | 16.6 | 5.2 | 12.8 | 11.0 | 11.1 | 11.7 |

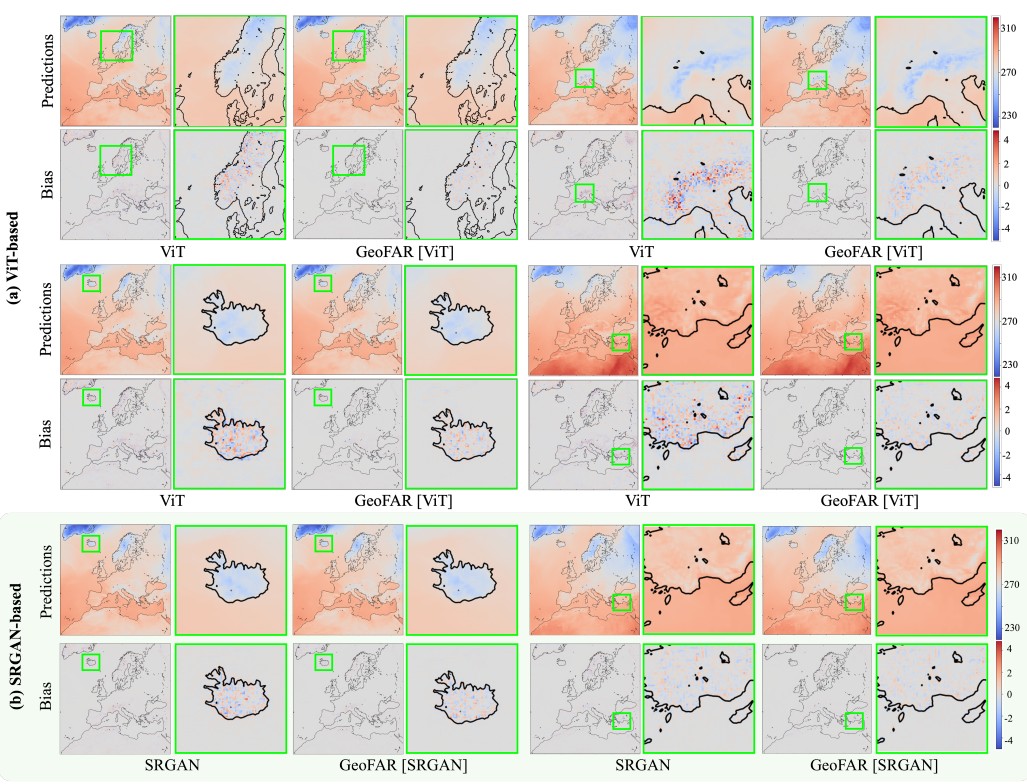

Figure 6: More examples of the super-resolution results on CERRA. (a) Results from ViT-based methods; (b) Results from SRGAN-based methods.

**Computational cost.** Table 9 compares the model size of baseline methods and our proposed method. Compared to baseline models like U-Net, our proposed approach only introduces 0.9M additional parameters and slightly slows down the performance by about 3 FPS. Even on a target image of 11 $km$ resolution ($534 \times 534$) covering the whole Europe, GeoFAR is able to run at over 11 FPS on a RTX A5500 GPU, offering a favorable trade-off between extra computation and accuracy gains.

**Physical fidelity.** We evaluate the physical fidelity of predicted wind fields (10u, 10v)) on CERRA $\times 2$. Table 10 reports the RMSE for the 10 m zonal and meridional winds, as well as the kinetic energy spectral RMSE ($RMSE_{KE}$, equation 18), for the ViT baseline and GeoFAR. GeoFAR substantially reduces the pointwise errors of both $10u$ and $10v$, and, more impor-

Table 10: Comparison of physical fidelity for 10 m winds using RMSE and kinetic energy spectral RMSE ($RMSE_{KE}$) on CERRA.

| Method | $RMSE_{10u}$ | $RMSE_{10v}$ | $RMSE_{KE}$ |
|---|---|---|---|
| ViT | 0.341 | 0.355 | 60.998 |
| GeoFAR[ViT] | 0.184 | 0.186 | 18.857 |

tantly, lowers $RMSE_{KE}$ from 60.998 to 18.857, indicating a much closer match to the target kinetic energy distribution and a higher physical fidelity of the reconstructed wind fields across the frequency components.

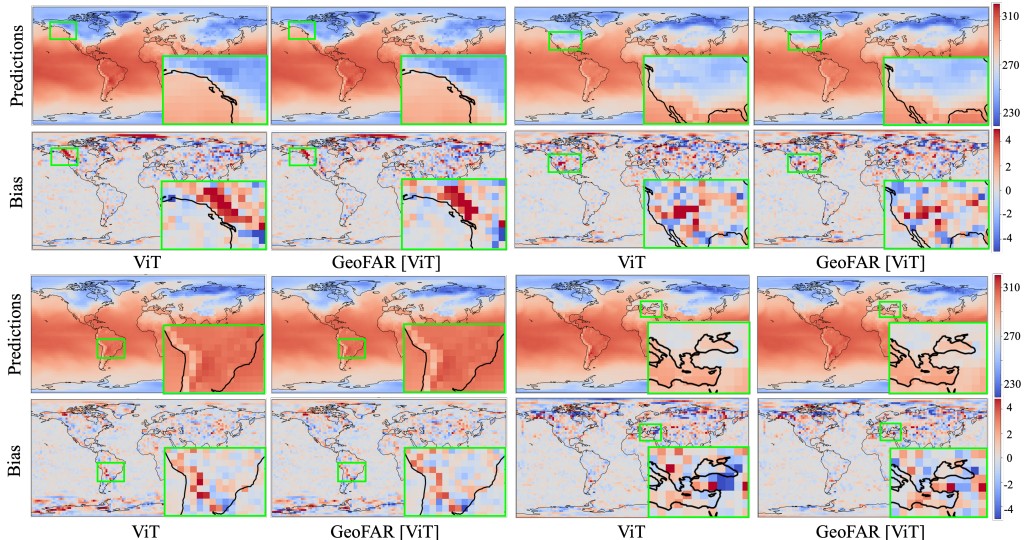

Figure 7: More examples of the super-resolution results on ERA5.

## A.8 VISUAL RESULTS

In this section, we provide more visualizations of the super-resolution results (covering both deterministic models and generative models), as well as visualizations of the terrain information.

**More results on CERRA and ERA5.** Figure 6 and Figure 7 provides more examples on the CERRA and ERA5 dataset, covering both deterministic methods (*i.e.,* ViT, GeoFAR[ViT]) and generative methods (*i.e.,* SRGAN, GeoFAR[SRGAN]). In these two figures, we provide more examples of different regions across the globe: from Iceland, the Scandinavian Peninsula, the Alps, to Anatolia, our proposed method consistently improves the fidelity and reduces the prediction bias in the super-resolved results of baseline models. Compared to deterministic models, generative baselines provide a better modeling of high-frequency information, as shown in Figure 6 (b). Nevertheless, the method proposed in this paper further alleviates artifacts and reduces the prediction bias caused by terrain on the basis of GAN. Moreover, we also note that the gains are generally more visible on CERRA (5.5 km) than on the much coarser ERA5 grid (2.8125°), where each grid aggregates information over larger areas. Nevertheless, even on ERA5, we still observe reduced bias in difficult regions with strong high-frequency variability, such as Alaska and the Balkan Peninsula (Figure 7).

**Frequency-aware representations.** Figure 8 provides some examples of the frequency-aware representations in different channels: the first/third row shows images convolved by FCK, and the second/fourth row shows the corresponding Fourier spectrum. As the frequency indices increase, the image shows increasing frequency, and the Fourier spectrum shows higher energy towards boundary regions. Compared to the DWT-decomposed subbands (Figure 9), which are mostly concentrated in the LL band, FAR shows a more fine-grained representation of frequency components and balanced energy distribution in different channels, which is crucial for the frequency-biased climate data.

**Terrain information.** Figure 10 shows the elevation map of the globe and Europe, which is used to get the terrain vectors in our Geo-informed representations.

## A.9 LIMITATIONS AND FUTURE WORKS

In this paper, we focused on learning frequency-aware patterns and geo-informed representations to for climate downscaling. A limitation is that our geographic factors mainly characterize the land surface, whereas atmospheric dynamics at multiple pressure levels also shape near-surface variability. Extending the framework to multi-dimensional implicit neural representations that are conditioned on pressure level, height, and key atmospheric variables (e.g., winds, humidity) is a natural next step. A second limitation is our single-variable setting: cross-variable relationships between variables (e.g., among temperature, humidity, pressure, precipitation, and wind) are not explicitly mod-

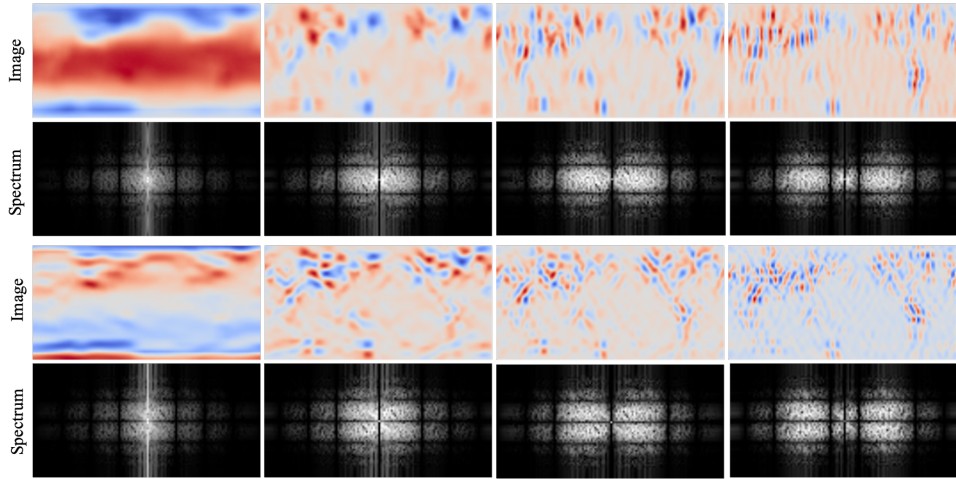

Figure 8: Examples of the first eight subbands from FAR out of 64 subbands (ERA5).

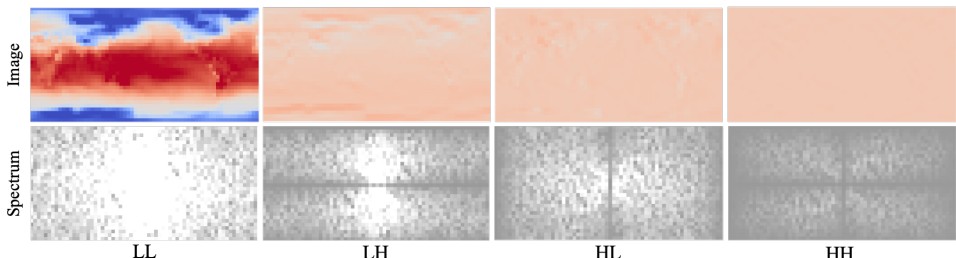

Figure 9: Examples of the decomposed subbands from DWT (ERA5).

eled. Future directions include multi-variable, physically consistent downscaling via shared latent fields and cross-variable operators, and the incorporation of soft or hard physical constraints (e.g., hydrostatic balance, thermodynamic identities) through physics-informed losses or differentiable solvers. More broadly, how to jointly capture cross-variable dependencies while enforcing physical consistency and providing calibrated uncertainty estimates remains an important open question.

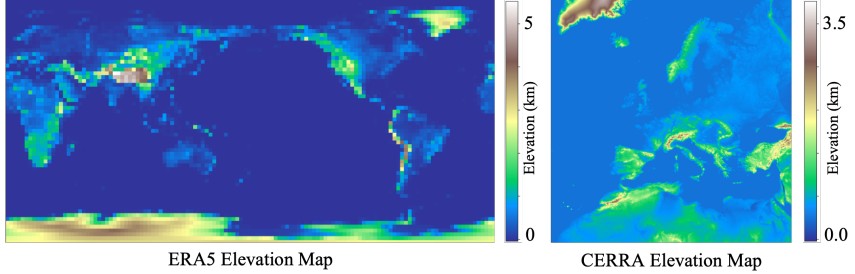

Figure 10: The elevation maps of ERA5 and CERRA.

### A.10 LLM USAGE

We used LLMs as assistive tools during the writing of this paper. Specifically, LLMs were employed for polishing grammar and improving clarity. LLMs were not involved in research ideation and experimental design.

