# OpenReview forum: "GeoFAR: Geography-Informed Frequency-Aware Super-Resolution for Climate Data"
_ICLR.cc/2026/Conference — ICLR 2026 Poster_

### Official Review · Reviewer_vEup · 2025-10-28

**Soundness:** 3
**Presentation:** 2
**Contribution:** 3
**Rating:** 6
**Confidence:** 3

**Summary:**

The authors argue that during image super-resolution using DNNs, DNNs tend to preserve low-frequency structures while neglecting the retention of high-frequency components. This leads to performance degradation when applying DNNs to meteorological data super-resolution tasks, which are predominantly low-frequency. To address this issue, the authors propose GeoFAR, a plug-and-play module. GeoFAR encodes input data into frequency-domain representations and additionally introduces implicitly encoded spatial information as input to the DNN, thereby improving performance on climate data super-resolution tasks dominated by low-frequency content. The authors validate the effectiveness of their method across multiple datasets and various network architectures.

**Strengths:**

1. GeoFAR is a plug-and-play method, and the authors demonstrate that it is better suited for meteorological data super-resolution tasks compared to commonly used encoding methods such as DWT.

2. GeoFAR shows performance improvements across multiple neural network architectures, suggesting that performing super-resolution in the frequency domain is a more suitable approach for meteorological data super-resolution, offering new insights to the community.

3. Implicitly encoding spatial information into features usable by neural networks is an important research direction in recent years, and the authors present a compelling practical application in the context of super-resolution.

**Weaknesses:**

Overall, I find this an interesting paper, but the experimental validation raises concerns:

1. Directly downsampling high-resolution data to low resolution to form LR-HR pairs is an unreasonable practice. Such interpolation retains excessive high-frequency information. Given that the authors do not provide a detailed description of dataset construction, I question the validity of their datasets. Retaining excessive high-frequency details in the LR data constitutes information leakage—could this be the actual reason why GeoFAR, which optimizes in the frequency domain, appears effective? As shown in Table 1, comparing Global-to-Local Downscaling and Local Downscaling, the UNet results indicate that when the task involves real low-resolution to high-resolution data (e.g., ERA5-PRISM), the improvement from GeoFAR is significantly smaller than in simulated LR-HR scenarios.

2. The current experimental results lack comprehensive validation. In quantitative remote sensing, it is common practice to use multiple independent data sources for cross-validation—for example, jointly using reanalysis data and actual remote sensing observations—rather than relying solely on reanalysis data.

3. The paper suffers from significant writing issues, including imbalanced content depth, insufficient discussion of key issues, and poor organization. For instance, the related work section merely lists numerous existing methods without establishing a clear causal chain between “existing approaches” and the “problems summarized at the end of the section,” nor does it validate certain claims. Specifically, in Section 2.2, the authors hastily assert that “current methods remain based on natural images while poorly mimicking the biased frequency distribution of climate data,” yet they fail to provide a mathematical explanation for why existing methods underperform on meteorological data, nor do they compare against recent frequency-domain-based super-resolution methods presented at major conferences (e.g., BF-STVSR, DiffFNO).

**Questions:**

1. Can the authors validate the accuracy of their downscaling model using observational data—for example, by comparing against ground meteorological station measurements (evaluating accuracy at station-corresponding grid locations), MODIS LST (compared with ERA5 SKT), and GPM IMERG precipitation data (compared with ERA5 precipitation)?

2. Can the authors provide additional details regarding dataset construction and validate the reasonableness of their approach? For instance, do the downsampled data exhibit similar high-frequency content and entropy to real-world low-resolution observations? To my knowledge, directly interpolating HR remote sensing data to LR resolution often retains excessive high-frequency details—a phenomenon easily verifiable by downsampling Sentinel-2 data to 500m and comparing it with MODIS data. The remote sensing community typically simulates LR data using multi-stage downsampling combined with smoothing. Could different dataset construction strategies lead to different conclusions?

3. The authors repeatedly emphasize in the related work that existing DNNs fail to preserve high-frequency information and produce overly smooth outputs. However, as shown in Table 1, most tested DNNs already yield reasonably good results, and GeoFAR does not demonstrate a fundamental improvement. Does this suggest the authors overstate the limitations of current methods?

4. GeoFAR essentially performs input-level data augmentation without modifying the internal architecture of the DNN. Will the injected high-frequency information still be forgotten by the network? Does GeoFAR truly address the core issue, or does it merely provide a superficial fix?

5. How does GeoFAR compare against methods that explicitly incorporate frequency-domain considerations at the operator level? What about the latest methods from CVPR or ICCV 2025? Could GeoFAR be combined with them for even greater gains? (This would help substantiate the claim that GeoFAR is truly plug-and-play.)

6. As an open-ended question: how well does GeoFAR perform on super-resolution of various remote sensing indices—for example, NDVI (from MODIS 500m to Landsat 30m)? Super-resolution of remote sensing indices is more amenable to cross-validation using independent high-resolution observational data.

7. Can the authors add a clear statement of contributions? Currently, the paper’s contributions remain vague.

---

> ### Author Response · Authors · 2025-11-25
>
> > Robustness to different ways of generating low resolution inputs
> >> W1: "Directly downsampling high-resolution data to low resolution to form LR-HR pairs is an unreasonable practice...."
>
> >> Q2: Can the authors provide additional details regarding dataset construction and validate the reasonableness of their approach? For instance, do the downsampled data exhibit similar high-frequency content and entropy to real-world low-resolution observations? ... Could different dataset construction strategies lead to different conclusions?"
>
> Thank you for your comments. Downsampling high-resolution data is a common practice in climate super-resolution papers [1, 2, 3]. But we understand the comment and will try here to better detail our data preparation approach, before presenting some new results on a new dataset following a different strategy (a CERRA-based dataset where we construct the low-resolution input image by multi-stage downsampling combined with smoothing).
>
> - In the paper, we work on global (ERA5: $5.625^\circ$ to ERA5: $2.8125^\circ$), global-to-local (ERA5: $2.8125^\circ$ to PRISM: $0.75^\circ$), and local (CERRA: 22km to 11km, 22km to 5.5km, and 44km to 5.5km) super-resolution tasks. **For ERA5**, we use the preprocessed data from WeatherBench [1] to ensure a consistent data setting for a fair comparison with the methods from the Climate-Learn benchmark [2]. Weatherbench uses *bilinear interpolation* to regrid from fine ($0.25^\circ$) to coarse grids ($5.625^\circ, 2.8125^\circ$). **For PRISM**, we also use the processing method suggested by ClimateLearn and perform *bilinear interpolation* from 4km to resolutions. **For CERRA**, we follow this data processing manner: given the source data of 5.5 km resolution with $1069\times1069$ grids, we perform *bilinear interpolation* and regrid it to a series of low-resolution variants: 11 km ($534\times 534$), 22 km ($267\times 267$), and 44 km ($133\times 133$). With these variants, we get the aforementioned three local settings of different downscaling ratios.
>
>     Current downscaling studies mostly use bilinear interpolation or average pooling [1, 2, 3]. Following these papers, we use bilinear interpolation to regrid the newly proposed CERRA dataset. Different from direct interpolation methods (e.g., nearest-neighbor interpolation), bilinear interpolation/average pooling **implicitly includes data smoothing processing**, because each new grid value is a weighted average of neighboring points, and in the continuous view, it’s equivalent to convolving with a small low-pass filter.
>
> - To more comprehensively evaluate the model’s performance under more smoothed inputs, we construct **a new dataset from the CERRA database** whose low-resolution input image (22 km, $267\times 267$) is obtained by *a multi-stage bilinear interpolation* (5.5 km→ 11 km→ 22km) *combined with explicit smoothing of 2D Gaussian low-pass filtering at each stage* ($\sigma=1$). We perform experiments with both the ViT baseline and our method to super-resolve this smoothed input to the target image of 11 km resolution. Experimental results are presented in Table I: compared to the results without Gaussian smoothing, both methods suffer from a slight decrease of performance (RMSE, ViT: 0.380→0.436, GeoFAR: 0.191→0.213), however, GeoFAR still significantly improves the baseline performance (RMSE: 0.436→0.213) and remains robust to the input degradation. Therefore, our conclusions in the main paper remain unchanged even when using over-smoothed inputs.
>
> Table I: Results on the local downscaling setting (22km to 11km) with Gaussian smoothed input
>
> | Method | RMSE (w/ smoothing) | RMSE (w/o smoothing) | MB (w/ smoothing) | MB (w/o smoothing) | LFD (w/ smoothing) | LFD (w/o smoothing) |
> | --- | --- | --- | --- | --- | --- | --- |
> | ViT | 0.436 | 0.380 | 0.071 | 0.033 | 10.824 | 10.496 |
> | GeoFAR | 0.213 | 0.191 | 0.003 | -0.001 | 9.466 | 9.245 |
>
> We have added more details on how we construct the data in the Appendix (Section A.2: Line 902), and supplemented the corresponding results into the Appendix (Section A.7: Line 1158, Table 7b: Line 1134) as an analysis of the model’s robustness on data smoothing.
>
> Refs:
>
> [1] WeatherBench: A Benchmark Data Set for Data-Driven Weather Forecasting, JAMES, 2020.
>
> [2] Climatelearn: Benchmarking machine learning for weather and climate modeling, NeurIPS, 2023.
>
> [3] Hard-Constrained Deep Learning for Climate Downscaling, JMLR, 2023.

---

> ### Author Response · Authors · 2025-11-25
>
> > Validation on diverse climate and remote sensing data sources
> >> W2: "The current experimental results lack comprehensive validation. In quantitative remote sensing, it is common practice to use multiple independent data sources for cross-validation—for example, jointly using reanalysis data and actual remote sensing observations—rather than relying solely on reanalysis data."
>
> >>Q1: "Can the authors validate the accuracy of their downscaling model using observational data—for example, by comparing against ground meteorological station measurements (evaluating accuracy at station-corresponding grid locations), MODIS LST (compared with ERA5 SKT), and GPM IMERG precipitation data (compared with ERA5 precipitation)?"
>
> Thank you for your suggestion. We agree that evaluating on multiple independent data sources is important to assess the generality of downscaling methods beyond a single reanalysis product. In fact, our study already includes such cross-source settings (ERA5→PRISM), and we further expanded the analysis with the experiments on ERA5→MODIS in the revised manuscript.
>
> - In addition to reanalysis data (ERA5, CERRA), we also downscaled “max daily temperature” **from the reanalysis data (ERA5) to real-world station observations (PRISM)** in the paper, which are two independent data sources. ERA5 is a reanalysis data source widely used in climate science. PRISM is a station-based data source that aggregates station observations over the US into image-like grids. PRISM also introduces real-world challenges such as *spatially sparse observations and missing values,* the meteorological value not always being available across the whole grid, leading to incomplete grids. We address this by applying a foreground mask so that the model focuses on observed regions both in training and evaluation.
>
>     Applying GeoFAR in this setting yields notable improvements over both deterministic and generative baselines in the Global-to-Local downscaling experiment (Table 1 in the paper): ViT (RMSE: 2.163→1.745), U-Net (RMSE: 1.501→1.468), SRGAN (1.718→1.561). This demonstrates that our method remains beneficial when the target is a station-based observational product rather than a reanalysis.
>
> - To further broaden the range of data sources, we construct a new dataset aiming at **downscaling the daily skin temperature from ERA5-Land to the MODIS land surface temperature, i.e., from a land-focused reanalysis to a satellite-based remote sensing product.** More specifically, we download both datasets over Switzerland, which is a very challenging region with complex atmospheric processes associated with the complex alpine topography. We preprocess ERA5-Land to 11 km resolution ($21\times 46$), and  MODIS to roughly 1.1 km  ($210\times 460$) via bilinear interpolation, leading to a challenging setting of $\times 10$ downscaling. The training set is from 2018 to 2022, and the testing set is 2023. During the dataset construction process, we also encountered several challenges : 1) similar to PRISM, MODIS also has missing values in some regions, mostly due to cloud cover and fog, so we again used a mask to exclude these regions during training and testing, 2) sometimes, MODIS values are not usable due to technical issues (https://asdc.larc.nasa.gov/news/terra-data-and-imagery-outage-starting-october-10th-2022), so we excluded the data between 11-10-2022 and 22-10-2022.
>
>     Results are presented in Table II: overall, this task is significantly harder than ERA5/CERRA due to large domain and resolution gaps between the input and output, leading to higher error compared to the results on other datasets like ERA5 and CERRA. Nonetheless, this dataset still benefits from the GeoFAR method, with RMSE and Mean Bias significantly reduced on both ViT and U-Net baselines.
>
>
> Table II: Results on downscaling the skin temperature from ERA5 (11 km) to MODIS (1.1 km)
>
> | Method | RMSE | Pearson | MB |
> | --- | --- | --- | --- |
> | ViT | 3.474 | 0.771 | 1.822 |
> | GeoFAR[ViT] | 3.266 | 0.766 | 0.309 |
> | Unet | 3.715 | 0.720 | 1.696 |
> | GeoFAR[Unet] | 2.718 | 0.844 | 0.717 |
>
> We have supplemented the corresponding results into the Appendix (Section A.7: Line 1168, Table 7c: Line 1146) to demonstrate our method’s generalization ability of cross-product downscaling. We also describe the MODIS dataset in Appendix (Section A.2: Line 910) and will release the corresponding data downloading and processing code for better reproducibility.

---

> ### Author Response · Authors · 2025-11-25
>
> > Writing issues:
> >> W3: The paper suffers from significant writing issues, including imbalanced content depth, insufficient discussion of key issues, and poor organization. For instance, the related work section merely lists numerous existing methods without establishing a clear causal chain between “existing approaches” and the “problems summarized at the end of the section,” nor does it validate certain claims.
>
> Thank you for the comment. To enhance the clarity of the paper, we have significantly revised the related work to have a clear connection between the current literature and their key issues, we have also added a summary of contributions in the introduction (Line 96). Here, we explain each claim in the related work and the corresponding validation methods in detail:
>
> - Climate downscaling (Section 2.1: Line 130)
>     - **Explanation:** Although there are works proposing deterministic or generative methods for climate downscaling, *existing studies did not explicitly investigate the geography-dependent frequency characteristics of climate downscaling*: there is a severe low-frequency bias in the frequency distribution of climate variables, the loss of high-frequency information is strongly tied to the geography. The lack of insight makes most existing downscaling works *struggle to solve the high-frequency loss issue by leveraging the domain information (e.g., geography)*.
>     - **Validation:** We compared GeoFAR with both the state-of-the-art generic SR methods and task-specific climate SR methods (including the suggested Fourier-domain-based methods and diffusion-based methods), *our proposed methods achieve the best performance among all.*
> - Frequency-aware modeling (Section 2.2: Line 140-156)
>     - **Explanation:** Classic wavelet-based approaches tend to provide *limited decoupling (as four bands) of high-frequency content* in climate data whose distribution is concentrated at low frequencies. On the other hand, learnable *frequency attention can further exacerbate the loss of high-frequency information* due to its frequency truncation and the neural network's inherent bias towards low frequency. A fine-grained and unbiased decoupling method tailored for climate is still missing.
>     - **Validation:** We *compared against the wavelet-based method* in the ablation study (Table 4: Line 441), and in Appendix Figure 9 (Line 1315) we visualize the corresponding wavelet-decomposed climate variables to illustrate the limited separation of high-frequency structures. We also compared against the frequency attention in the above discussion to validate the advantage of our fixed kernel learning.
> - Geography-informed learning (Section 2.3 Line 160)
>     - **Explanation:** Although geography is important for climate modeling, existing deep learning approaches have a limited utilization of this prior information. For example, existing deep learning approaches typically incorporate elevation only as an extra input channel. Recent work on implicit neural representations provides more faithful ways to encode geographic prior information, but these techniques have not yet been developed for climate downscaling. Moreover, climate states depend jointly on location and elevation information, making *existing location-only encoding methods fail to fully exploit the spatial information for climate downscaling.*
>     - **Validation:** to the best of our knowledge, we are the first work that develop geographical implicit neural representations for climate downscaling, we also demonstrate the advantage of Geo-INR over location-only embeddings or feature stacking strategies in ablation study (Table 4: Line 440)
>
> To establish a clearer connection between the existing approaches and their problems, we have revised each subsection in related work.

---

> > ### Comment · Reviewer_vEup · 2025-11-25
> >
> > Thank you for the substantial revisions. I am pleased to see a clear improvement in the overall quality of the paper. On the ERA5–MODIS land surface temperature downscaling task, GeoFAR indeed delivers significant gains, and I find these results quite convincing. The readability of the manuscript has also been notably enhanced. Therefore, I am considering increasing my score to 8.
> >
> > However, I believe the construction of the HR–LR data still deserves further discussion. From my own experience, while directly interpolating HR data to obtain LR inputs is a common practice, we do observe a significant domain gap in unguided super-resolution settings without ground-truth LR (e.g., training on high-resolution WorldView and applying the model to super-resolve Sentinel-2). Without sufficiently strong smoothing of the training HR data, it is difficult to realistically mimic the information loss in Sentinel-2 caused by sensor characteristics, illumination conditions, atmospheric effects, etc. I am in the process of uploading some illustrative figures to better demonstrate this issue.

---

> > > ### Author Response · Authors · 2025-12-02
> > >
> > > In addition, we evaluated downscaling performance under stronger levels of input smoothing. The dataset is from CERRA, whose low-resolution input image (22 km, $267\times 267$) is generated by *multi-stage bilinear interpolation* (5.5 km→ 11 km→ 22km) *combined with explicit smoothing of 2D Gaussian low-pass filtering at each stage*. As shown in Table IV: increasing the Gaussian smoothing degree ($\sigma$) degrades the performance of both ViT and GeoFAR. However, GeoFAR consistently improves the baseline performance under different levels of smoothing. Therefore, our conclusions in the main paper remain unchanged even with over-smoothed inputs.
> > >
> > > Table IV: Results on the local downscaling setting (22km to 11km) under different levels of Gaussian smoothing ($\sigma$ is the standard deviation of the Gaussian kernel).
> > >
> > > | Smoothing | None | None | None | $\sigma=1.0$ | $\sigma=1.0$ | $\sigma=1.0$ | $\sigma=2.0$ | $\sigma=2.0$ | $\sigma=2.0$ |
> > > | --- | --- | --- | --- | --- | --- | --- | --- | --- | --- |
> > > | Metric | RMSE  | MB  | LFD | RMSE | MB  | LFD  | RMSE  | MB  | LFD  |
> > > | ViT | 0.380 | 0.033 | 10.496 | 0.436 | 0.071 | 10.824 | 0.577 | -0.002 | 11.368 |
> > > | GeoFAR | 0.191 | -0.001 | 9.245 | 0.213 | 0.003 | 9.466 | 0.344 | 0.000 | 10.419 |

---

> ### Author Response · Authors · 2025-11-25
>
> > Q3: The authors repeatedly emphasize in the related work that existing DNNs fail to preserve high-frequency information and produce overly smooth outputs. However, as shown in Table 1, most tested DNNs already yield reasonably good results, and GeoFAR does not demonstrate a fundamental improvement. Does this suggest the authors overstate the limitations of current methods?
>
> Thank you for your comment. Indeed, some DNNs achieve reasonably good global scores, and in GeoFAR we aim at improving on their limitations in handling high frequency information in geographically complex terrains. We are by no means claiming that current methods "fail" at super-resolution.
>
> With respect to the results:
>
> - The results in Table 1 aggregate errors over all regions. Since low-frequency components cover a large proportion of the data, errors on high frequencies and over complex terrain are less appreciable in the aggregated metrics. This is why most tested DNNs yield seemingly good results.
> - But when we look at more frequency-aware and geography-aware analyses (Fig. 3 cd), the advantage of GeoFAR becomes much clearer.
>     - Figure 3c shows that GeoFAR consistently outperforms the baseline across all wavelet subbands, with particularly large improvements in the high-frequency HL/LH/HH bands, therefore directly addressing the oversmoothing issue.
>     - Figure 3d further indicates that GeoFAR yields fundamental improvements at high elevations (in regions over 3km elevation, RMSE significantly drops from 1.755 to 0.210), where high-frequency and topographically induced variability is strongest and oversmoothing is most problematic in current DNNs.
> - The qualitative examples in regions of complex geography also highlight these differences. In the Alps (Fig. 4a), GeoFAR better resolves sharp temperature contrasts between valleys and ridges and reduces regional biases. Similar behavior is observed in the Scandinavian archipelago (Fig. 6), where GeoFAR more accurately captures fine-scale coastal and island structures that baseline models tend to smooth out.
>
> We have revised the related work (Line 130, Line 140-156, Line 160) and softened claims to avoid potential overstatement.
>
> > Q4: GeoFAR essentially performs input-level data augmentation without modifying the internal architecture of the DNN. Will the injected high-frequency information still be forgotten by the network? Does GeoFAR truly address the core issue, or does it merely provide a superficial fix?
>
> Thank you for your question. Our empirical results show a *distribution shift to high frequency is visible in both input (Fig. 4a) and output (Fig. 4b),* which demonstrates a better retention of high-frequency information in our model. Also, visual results in complex terrains (Fig. 4a: the Alps, Fig. 6: the Scandinavian archipelago) also demonstrate that GeoFAR reproduces high-frequency temperature patterns between adjacent valleys and ridges and recovers local details, showing significantly reduced high-frequency bias. These improvements benefit from our two core designs:
>
> - **The frequency-aware representation** explicitly encodes frequency-aware components and resamples high frequency in the filter bank, exposing the model to stronger high-frequency content. This shifts the model’s reliance to high-frequency inputs, improving the reconstruction of relevant high-frequency structures in the output. Also, this improvement is demonstrated by the ablation in Table 4 (+FCK brings RMSE from 0.233 to 0.216(CERRA) and from 1.110 to 1.100(ERA5))*.*
> - We not only enhance the high-frequency representation from the input, but also **perform fine-grained geography-informed learning in the latent space at the target resolution** by leveraging geographical implicit neural representations (w.r.t. location and elevation). This allows GeoFAR to modulate the fine-scale climate variability at the target grid, further mitigating the high-frequency lost issue.

---

> ### Author Response · Authors · 2025-11-25
>
> > Comparison with methods that explicitly incorporate frequency-domain considerations at the operator level
> >> W3: nor do they compare against recent frequency-domain-based super-resolution methods presented at major conferences (e.g., BF-STVSR, DiffFNO).
>
> >> Q5: How does GeoFAR compare against methods that explicitly incorporate frequency-domain considerations at the operator level? What about the latest methods from CVPR or ICCV 2025? Could GeoFAR be combined with them for even greater gains? (This would help substantiate the claim that GeoFAR is truly plug-and-play.)
>
> Thank you for your comments. BF-STVSR [6] and DiffFNO [7] are interesting papers based on neural operators in the Fourier domain, which we have included in the related works (Line 128, 146). However, we find these two papers not applicable to our task. BF-STVSR is proposed for a fundamentally different task: spatial and temporal interpolation in Video data, which we believe does not apply to climate downscaling. DiffFNO is an interesting paper on image super-resolution, but its code has not yet been released.
>
> Instead, we supplemented some new *operator-based and diffusion-based methods that directly target climate/meteorological science* and for which implementations are available. In particular, we include a recent **FNO-based climate super-resolution model** [8] as a strong operator baseline, and we also evaluate against **state-of-the-art diffusion-based downscaling methods** (e.g., ClimateDiffuse [9], STVD [10]).
>
> We follow the same experimental setup as the paper and perform experiments on three representative datasets (ERA5, ERA5 to PRISM, CERRA). Results are presented in Table III: the diffusion-based models, which are better at solving high-frequency details, perform better than deterministic baselines like U-Net in the high-resolution CERRA setting; by contrast, Fourier neural operators, which enhance the frequency representation in the Fourier domain, perform better than diffusion baselines in global-scale and global-to-local settings. Based on the state-of-the-art DSFNO method, our method still improves the baseline performance by large margins in all setups.
>
> Table III: Results on more state-of-the-art methods across global (ERA5), global-to-local (ERA5-Prism), and local (CERRA) settings on the T2m.
>
>
>
> |  | ERA5 |  |  | ERA5-Prism |  |  |  | CERRA |  |  |
> | --- | --- | --- | --- | --- | --- | --- | --- | --- | --- | --- |
> | Method | RMSE | MB | LFD | RMSE | MB | Pearson | LFD | RMSE | MB | LFD |
> | STVD | 1.310 | -0.029 | 9.462 | 1.781 | -0.185 | 0.960 | 8.288 | 0.255 | -0.065 | 9.747 |
> | ClimateDiffuse | 1.451 | 0.005 | 9.679 | 2.279 | -0.097 | 0.955 | 8.582 | 0.265 | -0.011 | 9.858 |
> | DSFNO | 1.265 | 0.019 | 9.397 | 1.546 | -0.032 | 0.968 | 8.015 | 0.343 | 0.004 | 10.397 |
> | GeoFAR[DSFNO] | 1.126 | -0.003 | 9.160 | 1.474 | -0.121 | 0.971 | 7.904 | 0.190 | 0.000 | 9.234 |
>
> In the revised paper, we have added the diffusion-based and FNO-based experimental results into Table 1: Line 343-347.
>
> Refs:
>
> [6] BF-STVSR: B-Splines and Fourier-Best Friends for High Fidelity Spatial-Temporal Video Super-Resolution, CVPR, 2025.
>
> [7] DiffFNO: Diffusion Fourier Neural Operator, CVPR, 2025.
>
> [8] Fourier Neural Operators for Arbitrary Resolution Climate Data Downscaling, JMLR, 2024.
>
> [9] Generative Diffusion-based Downscaling for Climate, arxiv, 2024.
>
> [10] Precipitation Downscaling with Spatiotemporal Video Diffusion, NeurIPS, 2024.
>
> > Q6: As an open-ended question: how well does GeoFAR perform on super-resolution of various remote sensing indices—for example, NDVI (from MODIS 500m to Landsat 30m)? Super-resolution of remote sensing indices is more amenable to cross-validation using independent high-resolution observational data.
>
> Thank you for the interesting question. Remote sensing super-resolution is indeed a promising direction. Conceptually, GeoFAR is not restricted to meteorological variables: it operates on generic coordinate fields and could, in principle, be applied to remote sensing indices affected by location and elevation variability. For example, in complex terrain, orography and related gradients can influence vegetation patterns, so our key idea of encoding location and elevation could still provide useful inductive bias. However, at 30 m resolution, the dominant source of fine-scale variability in NDVI is often very local land-cover heterogeneity rather than large-scale geographic trends. In this case, the benefits of location-based encoding would be less pronounced than elevation-based embedding. Designing and evaluating such an extension would require a dedicated study and careful comparison with existing remote-sensing SR methods, which we consider an exciting direction for future work, but we think this is outside the scope of this climate/meteorological paper. We hope the reviewer will understand our point of view.

---

> ### Author Response · Authors · 2025-11-25
>
> > Can the authors add a clear statement of contributions? Currently, the paper’s contributions remain vague.
>
> Thank you for your suggestion. We have added a clear statement of contribution at the end of the introduction (Line 95) to improve clarity.

---

> ### Author Response · Authors · 2025-11-25
>
> Thank you so much for your prompt reply, your recognition of our revision, and for sharing your experience! We are very pleased to hear about your experience in processing the satellite-based remote sensing products for super-resolution. In particular, the mentioned “Without sufficiently strong smoothing of the training HR data, it is difficult to realistically mimic the information loss in Sentinel-2 caused by sensor characteristics, illumination conditions, atmospheric effects” is a very valuable phenomenon that we didn’t notice before, and it will help us to consider the smoothing effect when working on simulating satellite-based images.
>
> In our case, we think there are differences in the acquisition processes between *satellite images* and *climate data* (including reanalysis data: ERA5, CERRA, and observational data: PRISM) used in our paper, making them show different characteristics, and require different processing methods:
>
> - Reanalysis data are obtained by assimilating multi-source observations (e.g., ground station observations, satellite radiance, ground-based GNSS, radiosondes) which **reconstruct the real-world and physically consistent atmospheric state** on a grid, and station based observations **directly observe atmospheric states** from the ground. For example, in CERRA reanalysis data[1], surface-level variables (e.g., 2 m temperature, surface pressure) are strongly anchored by dense in-situ station networks (SYNOP and other national networks). Station observations directly measure physical variables (e.g. 2 meter temperature, humidity, surface pressure) using ground-based sensors, **which is less affected by illumination conditions and cloud or fog block**. This makes the climate data used in the paper less affected by the Sentinel-2-like smoothing issues in low-resolution data. In this context, simple interpolation is a reasonable way to approximate a lower-resolution real atmospheric state sampled at different locations
> - Climate data reflects the real-world atmospheric physics, **which are governed by the primitive equations and constrained by topography and land sea contrast**. For example, in the case of wind flows and heat transport. Introducing strong smoothing in the climate/atmospheric variables can further enlarge the gap between synthesized data and real-world climate fields, and also degrade the physical fidelity. Therefore, we use only bilinear interpolation to generate the low-resolution data to avoid introducing distortions, which aligns with previous climate-related studies.
>
> In short, our goal in simulating LR inputs is to **mimic coarser climate state of the same underlying physical field**, rather than **simulating the LR observation from different imaging instruments**. Finally, to avoid potential information leakage, we also synthesized a smoothed dataset with CERRA (in the first question) and evaluated our methods, where the conclusions drawn from our paper remain unchanged.
>
> [1] Cerra, the copernicus european regional reanalysis system, Quarterly Journal of the Royal Meteorological Society, 2024.

---

### Official Review · Reviewer_oWKs · 2025-10-29

**Soundness:** 4
**Presentation:** 3
**Contribution:** 3
**Rating:** 8
**Confidence:** 3

**Summary:**

GeoFAR: Geography-Informed Frequency-Aware Super-Resolution for Climate Data
In this paper, the authors introduce GeoFAR, a novel approach to climate data super-resolution that explicitly addresses the frequency bias of neural networks and the geography-dependent variability that is inherent to climate data. The paper integrates a Frequency-Aware Representation (FAR), which is a convolutional module leveraging Discrete Cosine Transform (DCT) bases that separate and encode high- and low-frequency information. This approach also uses a Geography-Informed Implicit Neural Representation (Geo-INR), which uses a coordinate-based representaiton that jointly encodes spatial and elevation information using spherical harmonics and SIREN networks. GeoFAR can be used with any deterministic or generative SR model, and the authors impressively demonstrate SOTA results across different spatial resolutions, atmospheric variables, and downscaling ratios.

**Strengths:**

* well-framed motivation (Figure 1 is strong), supported by innovative design using frequency- and geography-aware representations
* having a model-agnostic design makes this approach easily adaptable
* the authors include extensive experiments across datasets, resolutions, and atmospheric variables to demonstrate the improvements from their approach. there seems to be consistent improvement over baselines, and the work clearly demonstrates that GeoFAR helps with high-frequency fidelity and reduces bias.
* analysis is well-structured, with comprehensive ablation studies and frequency+elevation sensitivity studies (Figure 5 was particularly interesting)

**Weaknesses:**

* not a necessity, but it would make the dense notation and equations in  more intuitive with a diagram (it could even go in the supplementary materials)
* although the comparisons in the authors’ experiments are broad, there are few diffusion baselines included (SR3 from Saharia et al. 2023; Watt & Mansfield 2024) in quantitative analysis and other methods could be helpful to look into (PhIREGAN from Stengel et al. 2020), despite being mentioning that they are strong recent approaches
* while the quantitative analysis is very informative on the whole, it would be food to include statistical analysis with variance across seeds if this is possible
* to more robustly evaluate the physical fidelity of the predictions, consider using additional evaluation frameworks such as generating kinetic energy spectra from the wind fields
* section 4.4 has the beginnings of this, but the paper would benefit by a deeper discussion of how to physically interpret the frequency and geography components and how they related to known phenomena
* in the appendix, it would be good to include a brief discussion of computational cost/runtime scaling…efficiency is another dimension that’s important to consider in the practical deployability of these methods
* it would be helpful to cite the relevant frequency-aware transformer works and coordinate-based climate model papers; the authors may also need to reference some work in the neural operators space (see Li et al. 2021) because they also tackle spectral bias in physical modeling

**Questions:**

* are the reported quantitative results averaged over multiple training seeds, or based on a single run?
* can you elaborate on how the frequency-aware and geography-informed components interact during training? for instance, does the Geo-INR modulate the FAR in a spatially adaptive way tied to physical processes, or mainly improve general representation quality?
* this is more of an aside, but the paper shows results for temperature and geopotential. how well does GeoFAR generalize to other variables (e.g., precipitation, humidity) with different spatial-frequency characteristics?
* how sensitive is the model’s performance to the truncation degree (L) in the spherical harmonics encoding? have you explored trade-offs between model accuracy and computational cost?
* could you clarify whether Geo-INR can be pretrained or shared across datasets/variables, or must it be trained jointly with each SR model?
* is there a timeline or specific repository planned? are there any preprocessing scripts or model-specific configurations that will be included for reproducibility?

---

> ### Author Response · Authors · 2025-11-25
>
> > it would make the dense notation and equations in more intuitive with a diagram
>
> Thank you for your suggestion, we have added a summary of notations, mapping functions, and meteorological variables in diagrams in the appendix (Section A.1: Page 16-17).
>
> > although the comparisons in the authors’ experiments are broad, there are few diffusion baselines included (SR3 from Saharia et al. 2023; Watt & Mansfield 2024) in quantitative analysis and other methods could be helpful to look into (PhIREGAN from Stengel et al. 2020), despite being mentioning that they are strong recent approaches
>
> Thank you for your suggestion. To strengthen the comparison with more recent state-of-the-art models, we additionally compare with three representative approaches:  **diffusion-based generative modeling (with the STVD[1], ClimateDiffuse [2]), and  Fourier neural operators for arbitrary-scale super-resolution** **(DSFNO [3])**.
>
> We follow the same experimental setup as the paper and perform experiemts on three representative datasets (ERA5, ERA5 to PRISM, CERRA). Results are presented in Table I: the diffusion-based models, which are better at solving high-frequency details, perform better than deterministic baselines like U-Net in the high-resolution CERRA setting; by contrast, Fourier neural operators, which enhance the frequency representation in the Fourier domain, perform better than diffusion baselines in global-scale and global-to-local settings. Based on the state-of-the-art DSFNO method, our method still improves the baseline performance by large margins in all setups.
>
> Table I: Results on more state-of-the-art methods across global (ERA5), global-to-local (ERA5-Prism), and local (CERRA) settings on the T2m.
>
> |  |  | ERA5 |  |  |  | ERA5-Prism |  |  |  |  | CERRA |  |
> | --- | --- | --- | --- | --- | --- | --- | --- | --- | --- | --- | --- | --- |
> | Method | RMSE | MB | LFD |  | RMSE | MB | Pearson | LFD |  | RMSE | MB | LFD |
> | STVD | 1.310 | -0.029 | 9.462 |  | 1.781 | -0.185 | 0.960 | 8.288 |  | 0.255 | -0.065 | 9.747 |
> | ClimateDiffuse | 1.451 | 0.005 | 9.679 |  | 2.279 | -0.097 | 0.955 | 8.582 |  | 0.265 | -0.011 | 9.858 |
> | DSFNO | 1.265 | 0.019 | 9.397 |  | 1.546 | -0.032 | 0.968 | 8.015 |  | 0.343 | 0.004 | 10.397 |
> | GeoFAR[DSFNO] | 1.126 | -0.003 | 9.160 |  | 1.474 | -0.121 | 0.971 | 7.904 |  | 0.190 | 0.000 | 9.234 |
>
> In the revised paper, we have added the diffusion-based and FNO-based experimental results into Table 1: Line 343-347.
>
> Refs:
>
> [1] Precipitation Downscaling with Spatiotemporal Video Diffusion, NeurIPS, 2024.
>
> [2] Generative Diffusion-based Downscaling for Climate, arxiv, 2024.
>
> [3] Fourier Neural Operators for Arbitrary Resolution Climate Data Downscaling, JMLR, 2024.
>
> > while the quantitative analysis is very informative on the whole, it would be good to include statistical analysis with variance across seeds if this is possible
>
> In our paper, we use the seed_everything(0) from PyTorch Lightning to control the random seeds of Python, NumPy, and PyTorch, in order to make our experiments as reproducible as possible. To further assess the robustness of our results, we additionally trained GeoFAR with several independent random seeds and report the variance across runs. As shown in Table II, the performance is highly stable across different seeds on the CERRA T2m 2× downscaling task: RMSE remain effectively unchanged, the mean bias and LFD only fluctuates within ±0.002. This indicates that our reported improvements are relatively robust to initialization seeds.
>
> Table II: Analysis with variance across seeds on CERRA (T2m, 2$\times$)
>
> | Seed | RMSE | MB | LFD |
> | --- | --- | --- | --- |
> | Paper: 0 | 0.191 | -0.001 | 9.245 |
> | 23 | 0.191 | 0.000 | 9.243 |
> | 149 | 0.191 | 0.001 | 9.242 |
> | 71 | 0.191 | -0.002 | 9.244 |
>
> We have included the analysis of the performance variance across seeds in Appendix (Section A.7: Line 1132, Table 7a: Line 1134)

---

> ### Author Response · Authors · 2025-11-25
>
> > to more robustly evaluate the physical fidelity of the predictions, consider using additional evaluation frameworks such as generating kinetic energy spectra from the wind fields
>
> Thank you for the insightful comment. In addition to the standard pointwise metrics, we have implemented an evaluation of the **kinetic energy (KE) spectra** of the predicted wind fields, to more directly assess the physical fidelity across spatial scales.
>
> To this end, we first construct a multi-variable super-resolution dataset from the CERRA database containing wind fields: 2m_temperature (T2m), 2m_relative_humidity (Rh2m), surface_pressure (Sp), 10m_u_component_of_wind (10u), 10m_v_component_of_wind (10v). As in the experimental setup of our paper, the native 5.5 km data ($1069\times 1069$) is further preprocessed to grids of 11 ($534\times 534$), 22 ($267\times 267$) km with 3-hour updates for the $2\times$ super-resolution. The training set is from 2010 to 2017, and the val/test sets are from 2018 to 2021, leading to a large-scale multi-variable downscaling dataset of roughly ~200GB.
>
> On this dataset, we build a baseline by stacking variables into different channels of a ViT. We also add our GeoFAR model to this same ViT. In both methods, we super-resolve the different variables jointly. We follow the experimental setting in Section A.5 both for training and evaluation.
>
> We then evaluate the physical consistency of the predicted wind fields by comparing their **KE spectra** [4] to those of the ground truth. Given horizontal wind components $u(x,y)$ and $v(x,y)$, we define a KE spectra-based RMSE to calculate the difference between the target fields and the predicted field, whose detailed formula can be found in Appendix, Section A.3.3 (Line 995).
> This error directly measures how well the model reproduces the distribution of kinetic energy across spatial scales.
>
> Table III reports these metrics for the ViT baseline and GeoFAR on the CERRA 10 m winds. GeoFAR not only significantly reduces the pointwise RMSE of the 10u and 10v components, but also yields a much lower KE-spectral RMSE (from 60.998 to 18.857), indicating a significantly improved match to the target kinetic energy distribution and a high consistency to the physical fidelity.
>
> Table III: A comparison on the physical fidelity of wind fields using the RMSE between kinetic energy spectra.
>
> | Method | $RMSE_{10u}$ | $RMSE_{10v}$ | $RMSE_{ke}$ |
> | --- | --- | --- | --- |
> | ViT | 0.341 | 0.355 | 60.998 |
> | GeoFAR[ViT] | 0.184 | 0.186 | 18.857 |
>
> In the revised paper, we have supplemented the evaluation on physical fidelity in Appendix, including the metric in Section A.3.3 (Line 995), and the results in Section A.7: Line 1232. We will also release the corresponding code for the evaluation framework.
>
> Refs:
>
> [4] Wind kinetic energy climatology and effective resolution for the ERA5 reanalysis, Climate Dynamics, 2022.

---

> ### Author Response · Authors · 2025-11-25
>
> > section 4.4 has the beginnings of this, but the paper would benefit by a deeper discussion of how to physically interpret the frequency and geography components and how they related to known phenomena
>
> Thank you for your insights that help us better interpret the model’s physical fidelity. Below, we clarify how these components relate to established physical scales and geographical controls.
>
> **Frequency components and atmospheric scales.** In our method, different frequency bands correspond to different spatial scales of variability in the atmospheric fields. Empirically (Fig. 8), the *low-frequency components* capture large-scale, slowly varying structures (e.g., continent–ocean and meridional temperature gradients), while the *high-frequency components* are dominated by localized features such as sharp frontal zones, and topographically induced anomalies. In our model, the frequency-aware strategy explicitly exposes these bands to the network, therefore, the resulting super-resolved variables not only fit the large-scale background but also more effectively reconstruct the fine-scale features that are crucial for applications such as the detection of local extremes and impact assessment.
>
> **Geographical representation and orographic effects.** The geographical component (Geo-INR) learns embedding of location and elevation for the target grids. In practice, this embedding learns similar patterns in regions with similar geographic characteristics: e.g., mountain ranges, plateaus, coastal zones, and lowland plains display similar learned representations . This aligns well with known physical controls: orography and latitude strongly modulate near-surface temperature, surface pressure, and wind patterns. We observe, for example, that grid points over the Alps or other high-elevation areas learn distinct geographic embeddings compared to nearby lowlands, and that these embeddings correlate with stronger high-frequency variability in both temperature and wind. Thus, the geography component can be seen as a data-driven encoding of orographic and regional effects that are known to shape the structure of climate fields.
>
> In the revised paper, we provide a deeper discussion on the physical interpretation of both the frequency and geography components in Section 4.4 (Line 479, Line 510).
>
> > in the appendix, it would be good to include a brief discussion of computational cost/runtime scaling…efficiency is another dimension that’s important to consider in the practical deployability of these methods
>
> Thank you for your suggestion. In addition to the model parameters, we supplement the analysis on the runtime (frames per second, FPS) by comparing different super-resolution architectures. As shown in Table IV, the GeoFAR variants preserve a favorable efficiency compared to the corresponding baselines. Even on a target image of 11 km resolution ($534\times 534$), covering the whole Europe, GeoFAR is able to run at over 11 FPS, trading a modest increase in parameters for improved accuracy while still being much faster than classic SR networks like ResNet and EDSR.
>
> Table IV: A comparison of model size and runtime. All models are tested on a working station with RTX A5500 GPU, on the CERRA dataset with input image size $534\times534$. Here U, V, S denote U-Net, ViT, and SRGAN, respectively.
>
> | Model | ResNet | U-Net | ViT | SRGAN | EDSR | SwinIR | GeoFAR[U] | GeoFAR[V] | GeoFAR[S] |
> | --- | --- | --- | --- | --- | --- | --- | --- | --- | --- |
> | #Params | 8.3 | 11.6 | 2.6 | 6.1 | 9.0 | 2.3 | 12.5 | 3.5 | 7.0 |
> | Speed (FPS) | 3.8 | 14.1 | 15.0 | 16.6 | 5.2 | 12.8 | 11.0 | 11.1 | 11.7 |
>
> In the revised paper, we have supplemented the analysis of the running time in the Appendix (Section A.7, Line 1227, Table 9: Line 1193)

---

> ### Author Response · Authors · 2025-11-25
>
> > it would be helpful to cite the relevant frequency-aware transformer works and coordinate-based climate model papers; the authors may also need to reference some work in the neural operators space (see Li et al. 2021) because they also tackle spectral bias in physical modeling
>
> Frequency-aware transformers, coordinate-based climate models, and neural operators are indeed closely connected to our work, and we have expanded the related work section accordingly. Concretely, we now discuss papers from each category in our revised related works, including the *Frequency-aware Token Reduction for Efficient Vision Transformer* [5] which selectively preserves high-frequency tokens to mitigate oversmoothing in ViTs; Climode encodes the spatial and temporal coordinates for weather forecasting [6]; *Fourier Neural Operator for Parametric Partial Differential Equations* [7] that projects functions to the Fourier domain for continuous function to function mapping, and its adaptability to the climate downscaling with *Fourier Neural Operators for Arbitrary Resolution Climate Data Downscaling* [3].
>
> We have cited relevant papers in the related work ([5]: Line 142, [6]: Line 191, [7]: Line 141) and included the comparison with the DSFNO[3] in the experimental section (Table 1: Line 345).
>
> Refs:
>
> [3] Fourier Neural Operators for Arbitrary Resolution Climate Data Downscaling, JMLR, 2024.
>
> [5] Frequency-Aware Token Reduction for Efficient Vision Transformer, NeurIPS, 2025.
>
> [6] ClimODE: Climate and Weather Forecasting with Physics-informed Neural ODEs, ICLR, 2024.
>
> [7] Fourier Neural Operator for Parametric Partial Differential Equations, ICLR, 2021.
>
> > can you elaborate on how the frequency-aware and geography-informed components interact during training? for instance, does the Geo-INR modulate the FAR in a spatially adaptive way tied to physical processes, or mainly improve general representation quality?
>
> We perform a per-pixel multiplication between each channel and pixel of the frequency-aware representation and the geographical embeddings. As explained in Equation 1, the Geo-INR modulates the FAR in a spatially adaptive way. Since we did not explicitly incorporate physics prior (e.g., as loss terms) to constrain the learning process, we did not tie our learning to specific physical processes, while empirical results (Figure 4) demonstrate that our implicit learning promotes geographically consistent SR results and improves the physical fidelity (as discussed in C4).

---

> ### Author Response · Authors · 2025-11-25
>
> > this is more of an aside, but the paper shows results for temperature and geopotential. how well does GeoFAR generalize to other variables (e.g., precipitation, humidity) with different spatial-frequency characteristics?
>
> Based on the CERRA database, we *construct a super-resolution dataset of five climate variables*: 2m_temperature (T2m), 2m_relative_humidity (Rh2m), surface_pressure (Sp), 10m_u_component_of_wind (10u), 10m_v_component_of_wind (10v). As in the experimental setup of our paper, the native 5.5 km data ($1069\times 1069$) is further preprocessed to grids of 11 ($534\times 534$), 22 ($267\times 267$) km with 3-hour updates for the $2\times$ super-resolution, the training set is from 2010 to 2017, and the val/test set is from 2018 to 2021, leading to a large-scale multi-variable downscaling dataset of roughly 200GB.
>
> On this dataset, we built a baseline by stacking variables into different channels of ViT, and also our GeoFAR model based on the ViT. In both methods, we super-resolve different variables jointly. We follow the experimental setting in Section A.5 for training and evaluation. Results are showin in Table V, where we have several interesting observations:
>
> - **Multi-variable super-resolution does not lead to improvements on all variables** (e.g., T2m is worse than single-variable training). Due to the fundamentally different physical meaning and large domain gaps of different climate variables, super-resolving multiple variables in a shared model (without guidance or constraints) does not necessarily lead to an improved performance. A similar phenomenon can be found in previous studies on downscaling [8, 9]. Although multi-variable super-resolution is feasible, single-variable climate super-resolution is still an important direction in climate science as it is relatively simple, efficient, and also we don’t always have access to multiple meteorological observations at the same location.
> - **In the multi-variable setting, GeoFAR still brings clear gains.** When we replace the ViT with GeoFAR (using the same multi-variable input/output configuration), all variables show better performance, and for variables strongly tied to topography (e.g. surface pressure), the improvements are impressive (e.g. RMSE reduction from 277.719 to 47.922 in our experiments)
>
> Table V: Results on multi-variable climate downscaling with CERRA (22km to 11km).
>
> | Variable | T2m  |  |  | 10u |  |  | 10v |  |  | Rh2m |  |  | Sp |  |  |
> | --- | --- | --- | --- | --- | --- | --- | --- | --- | --- | --- | --- | --- | --- | --- | --- |
> | Method | RMSE | MB | LFD | RMSE | MB | LFD | RMSE | MB | LFD | RMSE | MB | LFD | RMSE | MB | LFD |
> | ViT | 0.457 | 0.033 | 10.966 | 0.341 | 0.002 | 10.395 | 0.355 | -0.011 | 10.472 | 1.799 | -0.012 | 13.708 | 277.719 | -11.007 | 23.808 |
> | GeoFAR[ViT] | 0.262 | 0.001 | 9.859 | 0.184 | -0.000 | 9.163 | 0.186 | 0.000 | 9.187 | 1.215 | -0.003 | 12.921 | 47.922 | 0.375 | 20.291 |
>
> In the revised paper, we have merged the multi-variable super-resolution setup into the experimental setup in Section 4.1 (Line 351), we supplemented the results as Table 2 (Line 378), a discussion of experimental results in *Results across atmospheric variables* (Line 410), as well as the details on the dataset (Appendix A.2: Line 904). We will release the code for the multi-variable  dataset downloading, processing, and results reproduction.
>
> [8] Multi-Variable Hard Physical Constraints for Climate Model Downscaling. Proceedings of the AAAI Symposium Series, 2024.
>
> [9] Vision Transformers for Multi-Variable Climate Downscaling: Emulating Regional Climate Models with a Shared Encoder and Multi-Decoder Architecture. arxiv, 2025.

---

> ### Author Response · Authors · 2025-11-25
>
> > how sensitive is the model’s performance to the truncation degree (L) in the spherical harmonics encoding? have you explored trade-offs between model accuracy and computational cost?
>
> We have conducted an ablation study to assess how sensitive GeoFAR is to the truncation degree L in the spherical harmonics encoding. Table VI given below reports the number of parameters, inference speed, and RMSE on CERRA T2m (2×) for $L \in \{5, 7, 9\}$. These results show that the performance is not highly sensitive to L within a specific range: increasing L from 5 to 9 yields only a modest RMSE reduction (0.192 → 0.190), while slightly increasing the parameter count and reducing inference speed by about 5%. In our method, we therefore fix L=7, which offers a good trade-off between accuracy and efficiency.
>
> Table VI: The model’s performance on different truncation degrees (L)
>
> | L | 5 | 7 | 9 |
> | --- | --- | --- | --- |
> | # Params | 3.4 | 3.5 | 3.9 |
> | Speed | 11.3 | 11.1 | 10.7 |
> | RMSE | 0.192 | 0.191 | 0.190 |
>
> In the revised paper, we have supplemented the analysis of the model’s accuracy and speed to the truncation degree in Appendix (Section A.7: Line 1175, Table 8a: Line 1146).
>
> > could you clarify whether Geo-INR can be pretrained or shared across datasets/variables, or must it be trained jointly with each SR model?
>
> Thank you for your insightful comment. In the current work, Geo-INR is trained for each target variable and is used in a single-variable downscaling setting for the main experiments. Meanwhile, the aforementioned multi-variable results (Table III) show that each Geo-INR super-resolved variable can jointly benefit from learning with multiple variables. Conceptually, geographical coordinates and orography (e.g., latitude, longitude, elevation) are common across climate datasets and variables, and all affect both global and regional climate states (even though in different ways). This makes the key idea of Geo-INR probably adaptable to the line of general geographical embedding study. For example, by learning a mapping from geographical coordinates to a wide range of climate variables, the model can acquire a *general implicit climate representation conditioned on geography* [10], which can benefit a broad set of climate-related tasks (e.g., forecasting, projection) across different datasets and variables that are strongly influenced by location and orographic effects.
>
> [10] SatCLIP: Global, General-Purpose Location Embeddings with Satellite Imagery, AAAI, 2024.
>
> > is there a timeline or specific repository planned? are there any preprocessing scripts or model-specific configurations that will be included for reproducibility?
>
> Absolutely! We will release the repository after the double-blind reviewing process (about February). The repository will include all components needed for reproducibility, including data download instructions, dataset construction and preprocessing scripts, as well as the full model implementation with training and evaluation configurations.

---

### Official Review · Reviewer_AjLx · 2025-11-01

**Soundness:** 3
**Presentation:** 3
**Contribution:** 2
**Rating:** 4
**Confidence:** 4

**Summary:**

The paper proposed GeoFAR to address the issue of overfitting to low-frequency components in climate data downscaling. GeoFAR explicitly encodes climatic patterns at different frequencies and use the location and elevation information. The model was evaluted in both deterministic and generative super-resolution models over different datasets with various spatial resolutions, atmospheric variables, and downscaling ratios.

**Strengths:**

1.	The paper addresses an important issue in climate downscaling that existing models are often biased toward low-frequency components.
2.	The paper conducts comprehensive experiments over different datasets, different downscaling ratios over different variables and the ablation experiments show the improvements brought by each component of the proposed approach.

**Weaknesses:**

1.	The proposed model appears to use a combination of geolocation aware encoding and frequency-aware convolution kernel, which are similar to existing ideas. As example, the novelty of the frequency-aware convolution kernel using 2D DCT with N×N grid instead of 4 zones is not clear. It seems 2D DCT as the kernel weight has also been used in computer vision models such as Harmonic Convolutional Networks. Fcanet also uses similar idea to take DCT as the weight.
Harmonic convolutional networks based on discrete cosine transform. Pattern Recognition.
Fcanet: Frequency channel attention networks. In Proceedings of the IEEE/CVF international conference on computer vision.

2.	Though the model has compared many baseline models such as generative and deterministic models, the baseline models are a little bit out of date. For generative models, including more newer super resolution models will be helpful, like: Precipitation downscaling with spatiotemporal video diffusion. Advances in Neural Information Processing Systems. 2024.

**Questions:**

Please better explain the novelty of paper in terms of the model structure design. What is the key contribution of the frequency-aware convolution kernel?

---

> ### Author Response · Authors · 2025-11-25
>
> > The novelty of the geolocation aware encoding and frequency-aware convolution kernel
> >> The proposed model appears to use a combination of geolocation aware encoding and frequency-aware convolution kernel, which are similar to existing ideas. As example, the novelty of the frequency-aware convolution kernel using 2D DCT with N×N grid instead of 4 zones is not clear. It seems 2D DCT as the kernel weight has also been used in computer vision models such as Harmonic Convolutional Networks. Fcanet also uses similar idea to take DCT as the weight. Harmonic convolutional networks based on discrete cosine transform. Pattern Recognition. Fcanet: Frequency channel attention networks. In Proceedings of the IEEE/CVF international conference on computer vision.
>
> >> Please better explain the novelty of paper in terms of the model structure design. What is the key contribution of the frequency-aware convolution kernel?
>
> Thank you for your question.
>
> **About the frequency-aware convolution:** our frequency-aware convolutional kernel differs from previous works in both design motivation and proposed solution. FcaNet and Harmonic convolutional networks are *attention mechanisms* that adaptively attend to frequency components beneficial for high-level perception tasks (classification and detection), they truncates the high-frequency components (FcaNet selects the first 16 low frequency components) and the learnable kernel can further adapt to the biased input data; by contrast, and motivated by the biased distribution of climate variables towards low frequencies, our frequency-aware convolution kernel acts like *frequency hard resampling:* with $N\times N$ (64) kernels of increasing frequencies, we enhance and resample high-frequency components into multiple channels with the bank of band-pass filters mostly focusing on high-frequency components. In the paper, we showed that our design performs better than Discrete Wavelet Transform (4 zones) with both ablation studies (Table 4) and visualization (Appendix: Figure 9) since the DWT decoupled climate is highly aggregated at the LL band. Here, we also compare with FcaNet by replacing the FCK layer with FcaLayer (Table I): our proposed approach leads to better super-resolution performance.
>
> **About the geolocation-aware encoding:** our geographical implicit neural representation differs from previous works, since it *jointly encodes location and terrain-specific implicit representations* to inform climate SR. Also, Geo-INR is *the first attempt to learn a geographical implicit neural representation for the climate SR* task. Previous studies on geographical implicit neural representation mostly *focus on mitigating the distortions on 2D-location encoding* [3, 4] and on improving the accuracy of feature extraction [5], and overlook the specificity of atmospheric variables: they show stronger high-frequency variability in complex terrains, which are unique both in terms of location and terrain information. Our ablations in Table 4 also show that our Geo-INR outperforms the state-of-the-art location-only encoding methods.
>
> The proposed components are closely integrated, with frequency-aware convolutional coding enhancing high-frequency information, while Geo-INR modulates at each frequency component (especially the high frequencies), therefore enabling the network to generate geographically consistent super-resolution results.
>
> Table I:  Comparison with learnable kernels on ERA5 (T2m)
>
> | Method | RMSE | MB | LFD |
> | --- | --- | --- | --- |
> | GeoFAR w/ FcaLayer | 1.111 | 0.005 | 9.129 |
> | GeoFAR | 1.099 | 0.001 | 9.117 |
>
> In the related work section, we have included these frequency attention papers [1, 2] and further clarified the difference between this work and previous works (Section 2.2: Line 147, Section 2.3: Line 193). In the method section, we have provided more details on the frequency-aware representation (Line 232) and geographical representation learning (Line 271).
>
> Refs:
>
> [1] Harmonic convolutional networks based on discrete cosine transform. Pattern Recognition, 2022
>
> [2] Fcanet: Frequency channel attention networks, ICCV, 2021.
>
> [3] Multi-scale representation learning for spatial feature distributions using grid cells, ICLR, 2020.
>
> [4] Geographic location encoding with spherical harmonics and sinusoidal representation networks, ICLR, 2024.
>
> [5] Implicit Neural Representations with Periodic Activation Functions, NeurIPS, 2020.

---

> ### Author Response · Authors · 2025-11-25
>
> > Though the model has compared many baseline models such as generative and deterministic models, the baseline models are a little bit out of date. For generative models, including more newer super resolution models will be helpful, like: Precipitation downscaling with spatiotemporal video diffusion. Advances in Neural Information Processing Systems. 2024.
>
> Thank you for your suggestion. To strengthen the comparison with more recent state-of-the-art models, we additionally compare with three representative approaches:  **diffusion-based generative modeling (with the suggested STVD [6], ClimateDiffuse [7]), and  Fourier neural operators for arbitrary-scale super-resolution** **(DSFNO [8])**.
>
> We follow the same experimental setup as the paper and perform experiemts on three representative datasets (ERA5, ERA5 to PRISM, CERRA). Results are presented in Table II: the diffusion-based models, which are better at solving high-frequency details, perform better than deterministic baselines like U-Net in the high-resolution CERRA setting; by contrast, Fourier neural operators, which enhance the frequency representation in the Fourier domain, perform better than diffusion baselines in global-scale and global-to-local settings. Based on the state-of-the-art DSFNO method, our method still improves the baseline performance by large margins in all setups.
>
> Table II: Results on more state-of-the-art methods across global (ERA5), global-to-local (ERA5-Prism), and local (CERRA) settings on the T2m.
>
> |  |  | ERA5 |  |  |  | ERA5-Prism |  |  |  |  | CERRA |  |
> | --- | --- | --- | --- | --- | --- | --- | --- | --- | --- | --- | --- | --- |
> | Method | RMSE | MB | LFD |  | RMSE | MB | Pearson | LFD |  | RMSE | MB | LFD |
> | STVD | 1.310 | -0.029 | 9.462 |  | 1.781 | -0.185 | 0.960 | 8.288 |  | 0.255 | -0.065 | 9.747 |
> | ClimateDiffuse | 1.451 | 0.005 | 9.679 |  | 2.279 | -0.097 | 0.955 | 8.582 |  | 0.265 | -0.011 | 9.858 |
> | DSFNO | 1.265 | 0.019 | 9.397 |  | 1.546 | -0.032 | 0.968 | 8.015 |  | 0.343 | 0.004 | 10.397 |
> | GeoFAR[DSFNO] | 1.126 | -0.003 | 9.160 |  | 1.474 | -0.121 | 0.971 | 7.904 |  | 0.190 | 0.000 | 9.234 |
>
> In the revised paper, we have added the diffusion-based and FNO-based experimental results into Table 1: Line 343-347.
>
> Refs:
>
> [6] Precipitation Downscaling with Spatiotemporal Video Diffusion, NeurIPS, 2024.
>
> [7] Generative Diffusion-based Downscaling for Climate, arxiv, 2024.
>
> [8] Fourier Neural Operators for Arbitrary Resolution Climate Data Downscaling, JMLR, 2024.

---

### Official Review · Reviewer_s9rN · 2025-11-01

**Soundness:** 2
**Presentation:** 2
**Contribution:** 2
**Rating:** 4
**Confidence:** 4

**Summary:**

The paper proposes GeoFAR, a new method for climate downscaling. GeoFAR resolves the oversmoothness issue of existing deep learning models by explicitly encoding input data at different frequencies and injecting the model with geographical neural representations for location and elevation. Experiments show that GeoFAR performs better than existing baselines and helps mitigate high-frequency prediction errors in both deterministic and generative models.

**Strengths:**

- The paper aims to address an important problem in climate downscaling: oversmoothness and bias towards low-frequency information.
- The paper is well-written and the proposed method is relatively simple (which is good).

**Weaknesses:**

I have two main concerns: the soundness of the proposed method and the significance of the empirical results.
- The two proposed components, explicit frequency-aware encoding and geographical representation learning, aim to exploit high frequencies in the input, but what I think is more important is the output or ground-truth. When we train a deep learning model with an MSE loss, for example, regardless of how the model processes the input data, it can still learn to over-optimize for the low-frequency information in the ground-truth, and thus will still predict oversmooth fields.
- In practice, since the lower-resolution input is often smoother, exploiting high-frequency information in the input itself seems like a minimal gain.
- Empirically, there are only minor improvements when using GeoFAR, especially when compared with the UNet baseline. In the ablation study, I believe that if the Unet architecture is used, the difference would be minimal.
- Qualitative results in Figures 4 and 7 also show almost identical predictions of the ViT model with or without GeoFAR.

**Questions:**

- The Frequency-aware Convolution Kernels are fixed convolution kernels. Can they be learned so that the model is more data-driven?
- Can the proposed method be applied to multi-variable data? The experiments trained separate models for each variable, but I believe it's beneficial to use multiple variables in the input, so the model can exploit interactions between them and make better predictions.

---

> ### Author Response · Authors · 2025-11-25
>
> > On the soundness of the explicit frequency-aware encoding and geographical representation learning
>
> Thank you for your question, which helps us improve the explanation of how both components work:
> - **On the frequency-aware encoding.**
>
>     In climate downscaling, the frequency bias is a joint result of both the inherent bias of the super-resolution neural network and the biased data (from our statistical analysis in Figure 1).  Prior work shows that such bias can be mitigated both on the **model side** (e.g., high-frequency–aware ViTs [1], periodic activations [2]) and on the **data side** by modifying the input. For example, AdvFrequency avoids frequency shortcuts by changing the dataset statistics in the Fourier domain [3], SimPSI augments the input data to better preserve frequency components in time series data [4].
>
>     Our method **follows this second data-side line**: by explicitly encoding frequency-aware components and resampling high frequency in the filter bank, we expose the model to stronger high-frequency content. This shifts the model’s reliance to high-frequency inputs, improving the reconstruction of relevant high-frequency structures in the output. **Empirically, this improvement is demonstrated** by the *ablation in Table 4 (+FCK brings RMSE from 0.233 to 0.216 (CERRA) and from 1.110 to 1.100 (ERA5))*, and *the distribution shift to high frequency is directly visible in Fig. 4a and Fig. 4b*. A similar observation is drawn from previous studies [5]: by changing the spectrum of the training inputs, the model’s frequency bias can be changed accordingly.
>
>     **Processing the target image or training objectives is also an interesting perspective** on addressing the frequency bias issue. For example, we implemented the method FFL[6] and FACL[7] in Table 1, which propose Fourier-based losses between the target and prediction, *while in the climate SR task they didn’t work well*. For climate data, how to process the ground-truth while avoiding introducing distortions and creating gaps between the climate SR target can be interesting research questions to explore.
>
> - **On geographical representation learning.**
>
>     Our module is an **implicit neural representation defined at the target resolution conditioned on a fine-grained geographic embedding** about location and terrain information (because we first resize the input to the target resolution, following the setup of the ClimateLearn benchmark). Thus, Geo-INR modulates the reconstruction of each target pixel across regions and elevations in the latent space, instead of processing the input data. This makes our model able to recover more high-frequency climate variability during super-resolution. The behaviour of *geographical representation learning is illustrated in Figure 5* of the paper, where the learned geographic embeddings are of high resolution that capture fine-scale and geography-dependent patterns.
>
>
> To improve clarity, we have revised the method section with more details on the frequency-aware representation (Line 230) and geographical representation learning (Line 271).
>
> Refs:
>
> [1] Improving Vision Transformers by Revisiting High-frequency Components, ECCV, 2022.
>
> [2] Implicit Neural Representations with Periodic Activation Functions, NeurIPS, 2020.
>
> [3] Towards Combating Frequency Simplicity-biased Learning for Domain Generalization, NeurIPS, 2024.
>
> [4] SimPSI: A Simple Strategy to Preserve Spectral Information in Time Series Data Augmentation, AAAI, 2024.
>
> [5] How Does Frequency Bias Affect the Robustness of Neural Image Classifiers against Common Corruption and Adversarial Perturbations, IJCAI, 2022.
>
> [6] Focal frequency loss for image reconstruction and synthesis, ICCV, 2021.
>
> [7] Beyond using L2 loss for skillful precipitation nowcasting, NeurIPS, 2024.

---

> ### Author Response · Authors · 2025-11-25
>
> > Gains by exploiting high-frequency information in the input
>
> Thank you for your comment. Before discussing the numerical gains, we would like to clarify that, in our proposed method, we **not only enhance the high-frequency representation from the input, but also perform fine-grained geography-informed learning at the target resolution** by leveraging geographical implicit neural representations. This allows GeoFAR to inject frequency-aware patterns into the model and modulate the fine-scale climate variability at the target grid.
>
> Now, with respect to the gains:
>
> - The ablation results show that both components lead to **consistent improvements over strong baselines**, as shown in Table 4 of the paper.
> - We also note that **the data resolution can affect the gains by leveraging geography domain knowledge**: each pixel of ERA5 aggregates data over hundreds of kilometers, the geographical effect and climate variability have been smoothed to some extent; therefore, our improvement on CERRA (5.5km resolution) is more significant than on the coarser ERA5 grid (2.8125 degree resolution). As the spatial resolution increases, GeoFAR can better exploit high-frequency input information and perform more accurate geography-informed implicit representation.
> - In addition to the improvement reflected in the overall metrics, we also performed an elevation-aware analysis in Section 4.4 (Fig. 4d): **our proposed method consistently reduces the RMSE from lowlands to high plateaus, with the most significant gains at high elevations**: in regions over 3km elevation, where high-frequency climate variability is most pronounced, RMSE significantly drops from 1.755 to 0.210.
>
> We have revised the method section with more details on how we enhance high-frequency reconstruction (Line 230, 271), and also clarified in the experimental results that the gains from geography-informed learning can be affected by data resolution (Line 406) and regional geography characteristics (Line 503).
>
> > Empirically, there are only minor improvements when using GeoFAR, especially when compared with the UNet baseline.
>
> We agree that U-Net is a strong baseline and widely used for super-resolution, which is why **we explicitly tested the performance of U-Net and whether GeoFAR still brings benefits on top of the U-Net** across all experimental settings. As seen from the comparative experiments (Section 4.2) in Table 1 and 3, when building based on the U-Net, GeoFAR[U-Net] exhibits significant improvements in RMSE and LFD (mostly state-of-the-art) across spatial resolutions, atmospheric variables, and downscaling ratios.
>
> To further address your concern, we **perform additional ablations** based on a strong residual learning baseline with U-Net. As shown in Table I, adding our components one after another yields non-trivial gains, with the improvement of each component over 2% and overall improvement over 10%.
>
> Table I: Ablation study on CERRA T2m downscaling ($\times2$) based on U-Net.
>
> | Method | RMSE | Mean Bias | LFD |
> | --- | --- | --- | --- |
> | Baseline | 0.213 | 0.003 | 9.453 |
> | + FCK | 0.201 (-5.6%) | -0.003 | 9.312 |
> | + 2D-INR | 0.184 (-8.5%) | -0.001 | 9.147 |
> | + Geo-INR | 0.180 (-2.2%) | 0.003 | 9.127 |
>
> Thus, even when the base architecture is U-Net, the gains from GeoFAR are over 10%. We also note that these stronger U-Net baselines come with higher computational cost and slower inference (Table 9), whereas the ViT-based model offers a more favorable accuracy and efficiency trade-off.

---

> ### Author Response · Authors · 2025-11-25
>
> > Qualitative results in Figures 4 and 7 also show almost identical predictions of the ViT model with or without GeoFAR.
>
> **Climate predictions show strong regional differences**: we agree that the *predictions seem similar at a global or continental scale*, but the qualitative *improvements are apparent on regions of complex topography* (usually dominated by high-frequency). For example, in the zoom-in view of the Alps (Fig. 4a), GeoFAR reproduces fine-grained and high-frequency temperature patterns between adjacent valleys and ridges and recovers local details, showing significantly reduced regional bias. Similarly, in the Scandinavian archipelago (Fig. 6), where complex terrain and near-polar distortions make the task particularly challenging, GeoFAR provides significantly more accurate patterns than the baseline model.
>
> **Data resolution also modulates how visible these improvements are**: *gains are generally more visible on CERRA* (5.5 km) *than on the much coarser ERA5 grid* ($2.8125^{\circ}$), where each grid aggregates information over large areas. Nevertheless, even on ERA5, we still observe reduced biases in difficult regions with strong high-frequency variability, such as Alaska (Fig. 5b) and the Balkan Peninsula (Fig. 7).
>
> The regional heterogeneity of climate super-resolution is **further supported by our elevation-aware analysis** in Section 4.4 (Fig. 4d): GeoFAR consistently reduces RMSE from lowlands to high plateaus, with the *largest improvements at high elevations*, where fine-scale geographic effects dominate.
>
> We have clarified in the experimental results and the appendix that the visibility of gains from our method can be affected by data resolution (Line 406, Appendix: Line 1274) and regional geography characteristics (Line 503).
>
> > The Frequency-aware Convolution Kernels are fixed convolution kernels. Can they be learned so that the model is more data-driven?
>
> Thank you for your question. In principle, the frequency-aware kernels could be made learnable. In this work, however, **we keep the frequency-aware convolutional kernels fixed to avoid reintroducing the same frequency bias problem we are trying to mitigate**. By fixing kernel weights, the convolution acts as a localized filter bank that decouples the input into multiple variants, each emphasizing a specific frequency band. Meanwhile, this bank contains many high-frequency sensitive kernels. *By repeatedly sampling high-frequency components in the data, the network is exposed to more high-frequency inputs*, which strengthens its reliance on these components for target reconstruction.
>
> If the frequency filters are fully learnable, the network can again over-adapt to low-frequency components that are easier to optimize, and overlook the high-frequency structure. **There is indeed a line of work that learns data-driven frequency-aware representations** by first decomposing into frequency components and then learning attention over them (e.g., FcaNet-style frequency channel attention [8, 9]). In response to your comment, we implemented an FcaNet layer to reweight frequency-aware feature channels and replaced our fixed layer with this learnable variant. The resulting model (Table II) showed slightly worse performance, suggesting that in our climate super-resolution setting, learning flexible frequency weights tends to drift back the attention toward low-frequency dominance.
>
> Different from high-level perception tasks (e.g., image classification and object detection), which are less sensitive to the loss of high-frequency, in climate downscaling, making frequency-aware kernels learnable without regulation may exacerbate the model’s bias towards low-frequency information that is easy to optimize. Nevertheless, exploring *regularized* learnable frequency kernels (e.g., with explicit constraints) is an interesting direction to investigate in future works.
>
> Table II: Comparison with learnable kernels on ERA5 (T2m)
>
> | Method | RMSE | MB | LFD |
> | --- | --- | --- | --- |
> | GeoFAR w/ FcaLayer | 1.111 | 0.005 | 9.129 |
> | GeoFAR  | 1.099 | 0.001 | 9.117 |
>
> In the revised paper, we have explained the details on fixed kernels and its benefits in both related work (Section 2.2: Line 147) and method sections (Section 3.1: Line 230-235).
>
> Refs:
>
> [8] Harmonic convolutional networks based on discrete cosine transform, Pattern Recognition, 2022.
>
> [9] FcaNet: Frequency Channel Attention Networks, ICCV, 2021.

---

> ### Author Response · Authors · 2025-11-25
>
> > Can the proposed method be applied to multi-variable data? The experiments trained separate models for each variable, but I believe it's beneficial to use multiple variables in the input, so the model can exploit interactions between them and make better predictions.
>
> Thank you for your question. Yes, **our proposed method can be applied to multi-variable climate downscaling and improve the performance.**
>
> Based on the CERRA database, we **construct a super-resolution dataset of five climate variables**: 2m_temperature (T2m), 2m_relative_humidity (Rh2m), surface_pressure (Sp), 10m_u_component_of_wind (10u), 10m_v_component_of_wind (10v). As in the experimental setup of our paper, the native 5.5 km data ($1069\times 1069$) is further preprocessed to grids of 11 ($534\times 534$), 22 ($267\times 267$) km with 3-hour updates for the $2\times$ super-resolution, the training set is from 2010 to 2017, and the val/test set is from 2018 to 2021, leading to *a large-scale multi-variable downscaling dataset of roughly 200GB*.
>
> On this dataset, we built a baseline by stacking variables into different channels of ViT, and also our GeoFAR model based on the ViT. In both methods, we super-resolve different variables jointly. We follow the experimental setting in Section A.5 for training and evaluation. Results are showin in Table III, where we have several interesting observations:
>
> - **Multi-variable super-resolution does not lead to improvements on all variables** (e.g., T2m is worse than single-variable training). Due to the fundamentally different physical meaning and large domain gaps of different climate variables, super-resolving multiple variables in a shared model (without guidance or constraints) does not necessarily lead to an improved performance. A similar phenomenon can be found in previous studies on downscaling [10, 11]. Although multi-variable super-resolution is feasible, single-variable climate super-resolution is still an important direction in climate science as it is relatively simple, efficient, and also we don’t always have access to multiple meteorological observations at the same location.
> - **In the multi-variable setting, GeoFAR still brings clear gains.** When we replace the ViT with GeoFAR (using the same multi-variable input/output configuration), all variables show better performance, and for variables strongly tied to topography (e.g. surface pressure), the improvements are impressive (e.g. RMSE reduction from 277.719 to 47.922 in our experiments)
>
> Table III: Results on multi-variable climate downscaling with CERRA (22km to 11km).
>
> | Variable | T2m  |  |  | 10u |  |  | 10v |  |  | Rh2m |  |  | Sp |  |  |
> | --- | --- | --- | --- | --- | --- | --- | --- | --- | --- | --- | --- | --- | --- | --- | --- |
> | Method | RMSE | MB | LFD | RMSE | MB | LFD | RMSE | MB | LFD | RMSE | MB | LFD | RMSE | MB | LFD |
> | ViT | 0.457 | 0.033 | 10.966 | 0.341 | 0.002 | 10.395 | 0.355 | -0.011 | 10.472 | 1.799 | -0.012 | 13.708 | 277.719 | -11.007 | 23.808 |
> | GeoFAR[ViT] | 0.262 | 0.001 | 9.859 | 0.184 | -0.000 | 9.163 | 0.186 | 0.000 | 9.187 | 1.215 | -0.003 | 12.921 | 47.922 | 0.375 | 20.291 |
>
> In the revised paper, we have merged the multi-variable super-resolution setup into the experimental setup in Section 4.1 (Line 351), we supplemented the results as Table 2 (Line 378), a discussion of experimental results in *Results across atmospheric variables* (Line 410), as well as the details on the dataset (Appendix A.2: Line 904). We will release the code for the multi-variable  dataset downloading, processing, and results reproduction.
>
> Refs:
>
> [10] Multi-Variable Hard Physical Constraints for Climate Model Downscaling. Proceedings of the AAAI Symposium Series*, 2024.*
>
> [11] Vision Transformers for Multi-Variable Climate Downscaling: Emulating Regional Climate Models with a Shared Encoder and Multi-Decoder Architecture. arxiv, 2025.

---

### Author Response · Authors · 2025-11-24

We thank the reviewers for their thoughtful feedback and for emphasizing the *importance of tackling oversmoothing in climate downscaling* (s9rN, AjLx). We are encouraged that GeoFAR is considered *simple but effective* (s9rN), *well-motivated with an innovative and model-agnostic design* (oWKs), and that *it offers new insights to the community* (vEup) in performing climate SR in the frequency domain. We are also glad that our experiments are found to be *comprehensive* (AjLx), with *consistent improvements across datasets and baselines*, *clearly demonstrating gains in high-frequency fidelity* (oWKs, vEup). Finally, we appreciate the remarks noting that the paper is *well-written* (s9rN), *with well-structured and interesting analysis* (oWKs). We believe that the remaining concerns can be well-resolved through our point-by-point responses during the discussion period. **A revised version of the paper and appendix, with all changes highlighted in blue, has been uploaded.** Please do not hesitate to reach out with any further questions.

---

### Author Response · Authors · 2025-12-02
**A summary of discussion**

Dear Reviewers and Area Chairs,

We would like to sincerely thank you for your efforts in handling our submission. Below is a summary of our responses:

**Response to reviewer** [s9rN](https://openreview.net/forum?id=0WHpOekph0&noteId=bsiiWTRQEX)**:**

- Clarified the method's soundness by adding previously missing details (w1, w2)
- Demonstrated the significance of improvements with additional UNet ablation (Table I) and geography-dependent gains (w3, w4)
- Compared with data-driven frequency-aware encoding (Table II), showing the benefits of our fixed kernel design (q1)
- Added experiments on a newly constructed multi-variable downscaling dataset (Table III); GeoFAR shows consistent improvements (q2)

**Response to reviewer** [AjLx]([https://openreview.net/forum?id=0WHpOekph0&noteId=bsiiWTRQEX](https://openreview.net/forum?id=0WHpOekph0&noteId=tqMJXq0WUb)):

- Clarified the difference from previous frequency encodings and compared GeoFAR with them (Table I), showing the benefits of the fixed kernel design (w1&q1)
- Compared with recent generative models ([STVD](https://proceedings.neurips.cc/paper_files/paper/2024/file/669b0bb64ff9eff7a81a7858a54fe7a0-Paper-Conference.pdf), [ClimateDiffuse](https://arxiv.org/html/2404.17752v1), and [DSFNO](https://www.jmlr.org/papers/volume25/23-0597/23-0597.pdf)); GeoFAR achieves stronger performance and further improves DSFNO (Table II) (w2)

**Response to reviewer** [oWKs](https://openreview.net/forum?id=0WHpOekph0&noteId=5omP8j9iOi):

- Improved writing with notation summary, physical interpretation, and more citations ([DSFNO](https://www.jmlr.org/papers/volume25/23-0597/23-0597.pdf), [Frequency-aware Transformer](https://openreview.net/pdf?id=Dr06Wjh45k), [FNO](https://openreview.net/forum?id=c8P9NQVtmnO)) (w1, w5&q2, w7)
- Compared with recent generative models ([STVD](https://proceedings.neurips.cc/paper_files/paper/2024/file/669b0bb64ff9eff7a81a7858a54fe7a0-Paper-Conference.pdf), [ClimateDiffuse](https://arxiv.org/html/2404.17752v1), and [DSFNO](https://www.jmlr.org/papers/volume25/23-0597/23-0597.pdf)); GeoFAR achieves stronger performance and further improves DSFNO (Table I) (w2)
- Analyzed robustness (random seeds, truncation degree) and physical fidelity (Tables II, III, VI; w3&q1, w4, q4).
- Evaluated the modest increase in parameters and runtime (Table IV) (w6)
- Confirmed the gains across more variables with multi-variable results (Table V) (q3)
- Confirmed the plan for code release (q6)

**Response to reviewer** [vEup](https://openreview.net/forum?id=0WHpOekph0&noteId=8FeIO8uyHv):

- Explained low-resolution input construction in each dataset, investigated robustness to data smoothing (Table I) (w1&q2)
- Clarified PRISM results and added ERA5–MODIS experiments, showing generalization to observational data (Table II) (w2&q1)
- Reorganized related work and added a summary of contributions (w3&q7)
- Explained the significant improvements in high-frequency bands and regions (q3&q4)
- Compared with neural operators: [DSFNO](https://www.jmlr.org/papers/volume25/23-0597/23-0597.pdf); GeoFAR shows better performance and further improves it (Table III) (q5)

As a result, reviewer vEup [acknowledged on 25-11-2025](https://openreview.net/forum?id=0WHpOekph0&noteId=MkJw1PYSkF) **the convincingness of the response and expressed the willingness to increase the score to 8,** with one remaining concern about *the construction of the HR–LR data.* To further address this:

- We clarified that the different acquisition processes of climate data and satellite images require different LR constructions, and additionally validated the gains from GeoFAR with strongly smoothed inputs (Table IV).

We hope that the area chair considers this context in evaluation.

Thank you!

---

### Meta-Review · Area_Chair_TVi6 · 2026-01-05

**Summary:**

The paper tackles an important problem in climate downscaling with AI models of over-smooth predictions and present a frequency and geometry aware approach to increase the performance of SR models in this space. The reviewers raised several concerns:
1. Efficacy of only modulating the input (which is LR) for the SR task
2. Weak baselines; baselines should include newer generative models.
3. Concerns on robustness, physical fidelity, and interpretation of the results
4. Concerns on the flow of writing and unclear contributions
5. Concerns on the corruption of the LR data with some HR information that could have made the task easier (major concern)
6. Concerns on the novelty of the design of the NN compared to prior works

Overall, I believe the authors have done a good job in responding to all the concerns. I believe concern 5. might be still outstanding issue. It might not be sufficient to claim that existing SR works downsample HR inputs and hence this is the right way to approach the problem. However, the authors did add an observational dataset, a stronger smoothing data construction approach to ensure minimal corruption, and have engaged with the reviewer positively. The authors should add an honest discussion about this in their revision, based on the reviewer comments.

The novelty of the paper might be slightly limited and performance gains are modest. However, the ideas are interesting with sufficient evidence for the value of their methods.

**Reviewer Concerns:**

See summary above. Apart from concern 5., the authors seem to have addressed most of the other asks. They have added newer suggested baselines with diffusion models, neural operators, and Unets, all demonstrating the value of their methods. They have revised the paper to read better with better contextualization of their methodology. They have clarified the differences between their work and prior works.

**Reviewer Scores:**

The scores stand 4, 4, 8, 8 (possibly, provided the authors discuss the issue appropriately). I believe the 4s may have increased to borderline accepts with more participation in the discussions.

---

### Decision · Program_Chairs · 2026-01-26

Accept (Poster)